# How Transformers Learn In-Context Recall Tasks? Optimality, Training Dynamics and Generalization

## Abstract

We study the approximation capabilities, convergence speeds and on-convergence behaviors of one-layer decoder-only transformers trained on in-context recall tasks – which requires to recognize the *positional* association between a pair of tokens from in-context examples. Existing theoretical results only focus on the in-context recall behavior of transformers after being trained for *one* gradient descent step. It remains unclear what is the on-convergence behavior of transformers being trained by gradient descent and how fast the convergence rate is. In addition, the generalization of transformers in one-step in-context recall has not been formally investigated. This work addresses these gaps. We first show that a class of transformers with either linear, ReLU or softmax attentions, is provably Bayes-optimal for an in-context recall task. When being trained with gradient descent, we show via a finite-sample analysis that the expected loss converges at linear rate to the Bayes risks. Moreover, we show that the trained transformers exhibit out-of-distribution (OOD) generalization, i.e., generalizing to samples outside of the population distribution. Our theoretical findings are further supported by extensive empirical validations, showing that *without* proper parameterization, standard one-layer transformer models surprisingly *fail* to generalize OOD after being trained by gradient descent.

## 1 Introduction

Large language models (LLMs) have shown impressive results in complex tasks that require some form of "reasoning" where classical models such as feed-forward networks seem to struggle. These reasoning tasks include, but are not limited to, generating coherent and plausible texts from a given context, language understanding, and mathematical reasoning (Brown et al., 2020; Achiam et al., 2023). At the heart of LLMs is the transformer architecture that features the attention mechanism (Vaswani et al., 2017). Transformers can process a long sequence of contexts and enable in-context reasoning via attention mechanisms. Despite remarkable empirical performance, the theoretical understanding of attention in reasoning tasks remains elusive, raising critical risk and safety issues when it comes to the widespread adoption of LLM technology (Bommasani et al., 2021; Belkin, 2024).

The literature has shown the usefulness of disentangling the behavior of complex models such as LLMs via controlled-setting tasks for which we understand the groundtruth behaviors (Allen-Zhu, 2024). For understanding reasoning in LLMs, one of the benchmark tasks that the literature has been recently embarked on is next-token prediction (NTP), wherein the tasks require a model to understand the context from a sentence to be able to predict the next token correctly. As a running example, consider the task of predicting the next token for the sentence "After talking to Bob about Anna, Charles gives her email address to [?]". A global bigram statistics would predict the next token to be "the" as the bigram "to the" naturally occurs in English with high frequency. However, if another person appears in the context, say Bob, then "Bob" is perhaps a better token prediction, even though the bigram "to Bob" is not a frequent bigram in the global context. In transformers, there have been strong (both empirically and theoretically) evidence suggesting that attention heads are responsible for in-context reasoning such as the in-context bigram "to Bob" (Wang et al., 2022) while feedforward layers seem to be responsible for storing global statistics or factual knowledge such as the global bigram "to the" (Geva et al., 2020; Meng et al., 2022; Bietti et al., 2023; Nichani et al., 2024).

However, a proper understanding of how such capabilities emerge during training is still lacking. For example, it is unclear how distributional associations such as "to the" and in-context reasoning such as "to Bob" are automatically assigned to feed-forward layers and self-attention layers by gradient descent without being explicitly forced to do so during training. Several initial efforts have shed insights into the above question (Chen et al., 2025; Bietti et al., 2023; Nichani et al., 2024). While they made an important first progress, we are still far from depicting the whole picture of how reasoning emerges in transformers. In particular, the existing theoretical results are limited to the reasoning behavior for just the first gradient steps or an infinite-sample setting, which do not reflect how we actually train transformers in practice.

In this paper, we narrow the gap above by deriving an effective, interpretable structures of transformers for in-context recall tasks. We will show how these structures emerge through gradient descent training on a class of parameterized one-layer transformers with linear, ReLU and softmax attentions. Our main contributions are as follows.

- In Section 2, we formally define a new in-context reasoning task (Definition 2.1), in which multiple query tokens can appear in a sentence and the output tokens can be noisy. This is a more difficult version of the in-context recall tasks in existing works (Bietti et al., 2023; Chen et al., 2025). Our new data model also enables a natural way to setup out-of-distribution (OOD) testing via a set of *neutral* tokens.

- Section 3 considers the noiseless setting. In Lemma 3.1, we present the first parameterization of one-layer transformers with linear and ReLU attentions that are provably optimal for this setting. We show that this parameterization mimics a human-like strategy for solving the task, and that it can be realized via gradient descent training (Theorem 3.2). Furthermore, we prove that the trained model directly generalizes to OOD sentences (Theorem 3.3), as well as gradient descent alone is not implicitly biased towards this parameterization (Theorem 3.4).

- Section 4 studies the optimality and convergence of models with softmax attentions for the noiseless setting. Lemma 4.1 exhibits how the structure for softmax attention can be constructed from observing the structure for linear and ReLU attentions. Via a two-phase analysis, our Theorem 4.2 shows that the loss converges *at a linear rate* to 0.

- Section 5 studies the noisy setting. Lemmas 5.1 and 5.2 first demonstrate the Bayes-optimality of one-layer transformers with linear, ReLU and softmax attentions. Next, Theorem 5.4 shows that by adapting our parameterizations from the noiseless setting to the noisy setting, one-layer transformers admit a finite-sample analysis that results in a PAC-style high-probability generalization bound. Moreover, Theorem 5.6 explains how attention layer and feed-forward layer may converge to perform different functionalities.

- Finally, Section 6 presents experimental results demonstrating the advantages of our parameterization over non-parameterized models. These results reveal the crucial role of expressive power and parameterization in achieving Bayes-optimality and OOD generalization.

## 1.1 RELATED WORK

Several works have analyzed transformers' training dynamics for in-context learning of linear regression and binary classification. Ahn et al. (2024) show a one-layer linear transformer that performs a preconditioned gradient step, with $L$ layers corresponding to $L$ steps at certain critical points. Mahankali et al. (2023) find that a one-layer linear transformer trained on noisy data mimics a single least-squares gradient step. Zhang et al. (2024) prove convergence to a global minimum under suitable initialization. Huang et al. (2024a) study gradient descent in softmax transformers learning linear functions. Cui et al. (2024) show that multi-head attention with large embeddings outperforms single-head variants. Cheng et al. (2023); Li et al. (2025) demonstrate the ability to emulate gradient descent and generalize with Chain-of-Thought on nonlinear transformers. Siyu et al. (2024); Shen et al. (2025) studies the training dynamics of (multi-head) softmax transformers for multi-task linear regression and in-context classification. Tarzanagh et al. (2023) show that self-attention optimization mirrors hard-margin SVMs, revealing the implicit bias of 1-layer transformers trained via gradient descent, and that over-parameterization aids global convergence. Ataee Tarzanagh et al. (2023) demonstrate that gradient descent on softmax attention converges to a max-margin separator distinguishing between locally optimal tokens. Building on this, Vasudeva et al. (2024) provide finite-sample analysis. Deora et al. (2023) offer optimization and generalization guarantees for training single-layer multi-head attention models under the NTK regime.

Recent works have also examined transformers' training dynamics for next-token prediction (NTP). Tian et al. (2023a) show that self-attention acts as a discriminative scanner, focusing on predictive tokens and down-weighting common ones. Tian et al. (2023b) analyze multilayer dynamics, while Li et al. (2024) find that gradient descent trains attention to learn an automaton via hard retrieval and soft composition. Thrampoulidis (2024) study the implicit bias of gradient descent in linear transformers. Huang et al. (2024b) provide finite-time analysis for a one-layer transformer on a synthetic NTP task, showing sublinear max-margin and linear cross-entropy convergence. Their setting assumes one-to-one token mapping, whereas we address a more general case allowing one-to-many mappings and prove generalization results for this broader task.

Our work also connects to recent views of transformer weight matrices—especially in embedding and feed-forward layers—as associative memories. Bietti et al. (2023) show that transformers store global bigrams and adapt to new context at different rates. Chen et al. (2025) find that feed-forward layers capture distributional associations, while attention supports in-context reasoning, attributing this to gradient noise (though only analyzing one gradient step). Nichani et al. (2024) theoretically analyze gradient flow in linear attention models on factual recall tasks.

## 2 PROBLEM SETUP

**Notations.** We use bold lowercase letters for vectors and bold uppercase letters for matrices. Let $N$ be the size of the vocabulary, and $\mathcal{V} = [N] := \{1, \ldots, N\}$ be the vocabulary itself. A token $y \in [N]$ is an element of the vocabulary. A sentence of length $H$ is a sequence of tokens denoted by $z_{1:H}$, where $z_h \in [N]$ is the $h$-th element of $z_{1:H}$. We use $C_{x,y} = \sum_{h=1}^{H-1} \mathbb{1}\{z_{h-1} = x, z_h = y\}$ to denote the number of times a bigram $(x, y)$ appear in a sentence. Generally, we will use "word" and "token" interchangeably throughout the paper, although we often use "word" to refer to an element of a sentence and "token" to refer to a specific type of elements of the vocabulary.

**Definition 2.1 (Data Model - In-context Recall Tasks).** We study a modified variant of the noiseless and noisy in-context reasoning tasks proposed in Bietti et al. (2023) and Chen et al. (2025), respectively. More specifically, we define the following two special, non-overlapping sets of tokens: a set of trigger tokens $\mathcal{Q} \subset [N]$ and a set of output tokens $\mathcal{O} \subset [N]$, where $\mathcal{O} \cap \mathcal{Q} = \emptyset$. A special "generic" noise token is defined by $\tau = N + 1$. The noise level is determined by a constant $\alpha \in [0, 1)$, where $\alpha = 0$ corresponds to the noiseless learning setting (Bietti et al., 2023) and $\alpha > 0$ corresponds to the noisy learning setting (Chen et al., 2025). In our model, a sentence $z_{1:H+1}$ is generated as follows:

- Sample a trigger word $q \sim \text{Unif}(\mathcal{Q})$ and an output word $y \sim \text{Unif}(\mathcal{O})$.
- Sample randomly (over an arbitrary distribution) $z_{1:H-1}$ from the set of sentences that satisfy the following four conditions: (I) there exists at least one bigram $(q, y)$ in the sentence, (II) $\tau$ may appear in a sentence only if $\alpha > 0$, in that case $\tau$ is always preceded by $q$, (III) all bigrams of the form $(q, x)$ take either $x = y$ or $x = \tau$, and (IV) if another token $q'$ is in the sentence, then it is followed by an output word $y' \in \mathcal{O}$.
- Fix $z_H = q$,
- Set $z_{H+1} = \tau$ with probability $\alpha$ and $z_{H+1} = y$ with probability $1 - \alpha$.

**Comparisons to existing works.** Compared to the existing task modes in Bietti et al. (2023); Chen et al. (2025), our task model offers several notable advantages. First, all sentences in our models must contain at least one (trigger token, output token) bigram, leading to a better signal-to-noise ratio. This allows us to avoid un-informative sentences that contain no useful signals for learning. Second, our task models are *agnostic* with respect to the distribution of the sentences. In other words, we do not impose any assumptions on how words and sentences are distributed, as long as the conditions are satisfied. Thus, our distributionally agnostic models are both more applicable to practical scenarios and more challenging for theoretical analyses. Third, by restricting the output tokens to a subset of $[N]$, we can study the OOD generalization ability of a model on *unseen* output tokens. Note that existing work by Bietti et al. (2023); Chen et al. (2025) did not study OOD generalization. Furthermore, because there are more than one possible next-token for every trigger word, our next-token prediction task is more challenging than the task of learning a one-to-one token mapping in Huang et al. (2024b). We provide additional examples of real-life sentences in Appendix A.

**One-layer Decoder-only Transformers.** To establish the theoretical guarantees of the optimality and on-convergence behaviors of transformers, we adopt the popular approach in existing works (Bietti et al., 2023; Chen et al., 2025; Huang et al., 2024a, e.g.) and consider the following one-layer transformer model, which is a variant of the model proposed in Chen et al. (2025). Let $E : [N] \mapsto \mathbb{R}^d$

be the input word embedding, i.e. $E(z) \in \mathbb{R}^d$ is the input embedding of the word $z \in [N]$, and $\tilde{E} : [N] \mapsto \mathbb{R}^d$ be a (different) embedding representing the previous token head construction as in Bietti et al. (2023). Similar to the majority of existing works in the literature (Chen et al., 2025, e.g.), we employ a common assumption that the embeddings $E$ and $\tilde{E}$ are fixed and orthogonal, i.e. $E(i)^\top E(j) = \tilde{E}(i)^\top \tilde{E}(j) = \mathbb{1}\{i = j\}$ and $E(i)^\top \tilde{E}(j) = 0$ for any $i, j \in [N]$.

Let $U \in \mathbb{R}^{N \times d}, V \in \mathbb{R}^{d \times d}, W \in \mathbb{R}^{d \times d}$ be the unembedding matrix, the value matrix, and the joint query-key matrix, respectively. Our model consists of one attention layer and one feed-forward layer. The input $\mathbf{x}_{1:H}$ and the output of the model are as below.

$$\mathbf{x}_h := E(z_h) + \tilde{E}(z_{h-1}) \in \mathbb{R}^d,$$

$$\phi(\boldsymbol{x}_H, \boldsymbol{x}_{1:H}) := V \sum_{h=1}^{H} \sigma(\boldsymbol{x}_H^\top W \boldsymbol{x}_h) \boldsymbol{x}_h \in \mathbb{R}^d,$$

$$\xi_A := U\phi(\boldsymbol{x}_H, \boldsymbol{x}_{1:H}) \in \mathbb{R}^N, \tag{1}$$

$$\xi_F := U F (\boldsymbol{x}_H + \phi(\boldsymbol{x}_H, \boldsymbol{x}_{1:H})) \in \mathbb{R}^N,$$

where $F$ is the matrix of the linear layer and $\sigma : \mathbb{R} \to \mathbb{R}$ is the activation function which determines the range of the attention scores. For theoretical and empirical analyses, we use linear attention $\sigma(\boldsymbol{x}_H^\top W \boldsymbol{x}_h) = \boldsymbol{x}_H^\top W \boldsymbol{x}_h$, ReLU attention $\sigma(\boldsymbol{x}_H^\top W \boldsymbol{x}_h) = \max(0, \boldsymbol{x}_H^\top W \boldsymbol{x}_h)$ and softmax attention $\sigma(\boldsymbol{x}_H^\top W \boldsymbol{x}_h) = \frac{\exp(\boldsymbol{x}_H^\top W \boldsymbol{x}_h)}{\sum_{j=1}^{H} \exp(\boldsymbol{x}_H^\top W \boldsymbol{x}_j)}$. The final logit is $\xi = \xi_A + \xi_F$.

Compared to Chen et al. (2025), our model in (1) differs in the computation of $\xi_F$. More specifically, while their theoretical model used $\xi_F = U F \boldsymbol{x}_H$, we add $\sum_{h=1}^{H} \sigma(x_H^\top W \boldsymbol{x}_h) V \boldsymbol{x}_h$ to the input of the feed-forward layer, which is closer to the empirical model used for the experiments in Chen et al. (2025). As we will show in Section 5, this modification is sufficient for showing the Bayes-optimality of one-layer transformers. Next, similar to Chen et al. (2025), we fix the three embedding maps $E, \tilde{E}, U$, and use cross-entropy loss on $\xi$, i.e. the population loss is $L = \mathbb{E}_{q,y,\boldsymbol{z}} \left[ - \ln \frac{\exp(\xi_y)}{\sum_{j \in [N]} \exp(\xi_j)} \right]$.

## 3 WARMUP: NOISELESS LEARNING WITH LINEAR AND RELU ATTENTIONS

In this section, we consider the noiseless learning setting in which $\alpha = 0$ and $\tau$ never appears in a sentence. In Section 3.1, we prove the approximation capability of the model defined in (1) by showing that with linear and ReLU attentions, there exists a reparameterization of $U, V, F$ and $W$ that drives the population loss $L$ to 0. In Section 3.2, we show that the reparameterized model can be trained by normalized gradient descent (NGD) and the population loss converges to 0 at linear rate.

### 3.1 APPROXIMATION CAPABILITIES OF TRANSFORMERS ON NOISELESS SETTING

We show that for any instance of the noiseless data model in Definition 2.1, there is a one-layer transformer that precisely approximates the task instance, i.e., the population loss is zero. To this end, we initialize and freeze the matrix $F = 0$ so that $\xi_F = 0$. The population loss becomes

$$L(V, W) = \mathbb{E}_{q,y,\boldsymbol{z}} \left[ - \ln \frac{e^{\xi_{A,y}}}{\sum_{j \in [N]} e^{\xi_{A,j}}} \right] = \mathbb{E}_{q,y,\boldsymbol{z}} \left[ - \ln \frac{e^{\boldsymbol{e}_y^\top U V \sum_{h=1}^{H} \sigma(\boldsymbol{x}_H^\top W \boldsymbol{x}_h) \boldsymbol{x}_h}}{\sum_{j \in [N]} e^{\boldsymbol{e}_j^\top U V \sum_{h=1}^{H} \sigma(\boldsymbol{x}_H^\top W \boldsymbol{x}_h) \boldsymbol{x}_h}} \right], \tag{2}$$

where $\boldsymbol{e}_j$ is the $j$-th vector in the canonical basis of $\mathbb{R}^N$ (i.e., $[\boldsymbol{e}_j]_k = \mathbb{1}\{j = k\}$). Note that the output embedding $U$ is considered a fixed matrix as in Chen et al. (2025), thus the population loss is a function of $V$ and $W$. We consider a specific parametric class of the weight matrices $U, V$ and $W$. Studying a particular parametric class is a common approach for overcoming the highly non-convex landscape of transformers (e.g. Ahn et al., 2024; Yang et al., 2024; Huang et al., 2025a). In particular, the following lemma shows that there exists a reparameterization of $U, V$, and $W$ that makes the population loss arbitrarily close to 0. The proof is in Appendix D.1.

**Lemma 3.1.** *Let* $\boldsymbol{\lambda} = \{\lambda_k \in \mathbb{R}_+ : k \in \mathcal{Q}\}$ *be a set of* $|\mathcal{Q}|$ *of non-negative values. By setting* $U = [E(1) \ E(2) \ \dots \ E(N)]^\top, V = \boldsymbol{I}_d$ *and* $W = \sum_{k \in \mathcal{Q}} \lambda_k E(k) \tilde{E}^\top(k)$*, for both linear and ReLU attention, we obtain* $\lim_{\boldsymbol{\lambda} \to \infty} L(\boldsymbol{\lambda}) := \lim_{(\lambda_q)_{q \in \mathcal{Q}} \to \infty} L(V, W) = 0$.

*Proof.* (Sketch) The crucial observation is that the attention score of the $h$-th token $z_h$ is $\sigma(\boldsymbol{x}_H^\top W \boldsymbol{x}_h) = \lambda_q \mathbb{1}\{z_{h-1} = q\}$, which is a non-zero (and positive) value for $x_h$ only when $z_{h-1}$ is

the trigger word. As a result, the logits of token $j \in [N]$ is $\xi_j = \xi_{A,j} = \lambda_q C_{q,y} \mathbb{1}\{j = y\}$. Hence, the probability of outputting $y$ is $\lim_{\lambda_q \to \infty} \frac{\exp(\xi_y)}{\sum_{j \in [N]} \exp(\xi_j)} = \lim_{\lambda_q \to \infty} \frac{\exp(C_{q,y}\lambda_q)}{\exp(C_{q,y}\lambda_q) + N - 1} = 1.$ □

### 3.2 Convergence rate, Generalization and Implicit Bias of Gradient Descent

We analyze the dynamics of normalized gradient descent (NGD) in training one-layer transformers parameterized in Section 3.1. We will show that with linear and ReLU attentions, the population loss converges *linearly* to zero. Moreover, the trained model generalizes to samples that lie completely outside of the training population.

**Convergence rate of NGD.** From the proof sketch of Lemma 3.1, the population loss $L(\boldsymbol{\lambda})$ is $L(\boldsymbol{\lambda}) = \mathbb{E}_{q,y,z}\left[-\ln \frac{\exp(C_{q,y}\lambda_q)}{\exp(C_{q,y}\lambda_q) + N - 1}\right]$. We initialize $\boldsymbol{\lambda}_0 = \mathbf{0}$. Running standard gradient descent $\boldsymbol{\lambda}_{t+1} = \boldsymbol{\lambda}_t - \eta\nabla_{\boldsymbol{\lambda}}L$, where $\eta > 0$ is the learning rate, would require knowing the exact distribution of $z$ since $C_{q,y}$ is a random variable depending on $z$. Instead, we adopt an NGD algorithm, where

$$\boldsymbol{\lambda}_{t+1} = \boldsymbol{\lambda}_t - \eta\frac{\nabla_{\boldsymbol{\lambda}}L}{\|\nabla_{\boldsymbol{\lambda}}L\|_2}, \tag{3}$$

i.e. the gradient vectors are normalized by their Euclidean norm. A similar NGD update was used in Huang et al. (2024b) for learning an injective map on the vocabulary. The following theorem shows that the update (3) can be implemented without the knowledge of the distribution of $z$. Moreover, the population loss converges to zero at a linear rate. The proof can be found in Appendix D.2.

**Theorem 3.2.** *Starting from $\lambda_{q,0} = 0$ for all $q \in \mathcal{Q}$, the update rule (3) is equivalent to $\lambda_{q,t} = \frac{\eta t}{|\mathcal{Q}|}$ for all $t \geq 1$. Moreover, $L(\boldsymbol{\lambda}_t) = O(N\exp(-\eta t/|\mathcal{Q}|))$.*

**OOD Generalization to unseen output words.** The human-like strategy for solving the noiseless data model in Definition 2.1 is to predict the word that comes after a trigger token. That is, the position of the trigger token is the only important factor in this task. Such a strategy does not depend on what the actual output word $y$ is, and hence would easily generalize to a out-of-distribution sentence where the bigram $(q, y)$ is replaced by a new bigram $(q, y_{\text{test}})$, where $y_{\text{test}}$ is a non-trigger non-output word. The following theorem formalizes this intuition, indicating that our parameterization is precisely implementing this human-like strategy.

**Theorem 3.3.** *Fix any $y_{\text{test}} \in [N] \setminus (\mathcal{O} \cup \mathcal{Q})$. Take any **test** sentence generated by the noiseless data model, except that every bigram $(q, y)$ is replaced with $(q, y_{\text{test}})$. Then, our model after being trained by normalized gradient descent for $t$ steps, predicts $y_{\text{test}}$ with probability*

$$\Pr[z_{H+1} = y_{\text{test}} \mid \boldsymbol{\lambda}_t] := \frac{\exp(\xi_{y_{\text{test}}})}{\sum_{j \in [N]} \exp(\xi_j)} \geq \frac{\exp(\eta t/|Q|)}{\exp(\eta t/|Q|) + N - 1}.$$

*In particular, this implies that $\lim_{t \to \infty} \Pr[z_{H+1} = y_{\text{test}} \mid \boldsymbol{\lambda}_t] = 1$.*

**Reparameterization versus Directional Convergence.** In addition to the convergence of the loss function, existing works (e.g. Ji and Telgarsky, 2021; Huang et al., 2024b) on the training dynamics of neural networks learned with cross-entropy loss have shown that the trainable matrices *directionally convergence* to an optimal solution. More formally, a sequence of $\boldsymbol{A}_t$ directionally converges to some $\boldsymbol{A}_*$ if $\lim_{t \to \infty}\langle\frac{\boldsymbol{A}_t}{\|\boldsymbol{A}_t\|}, \frac{\boldsymbol{A}_*}{\|\boldsymbol{A}_*\|}\rangle = 1$. In our work, the joint query-key matrix $\boldsymbol{W}$ is reparameterized as a form of associative memory of the trigger tokens, rather than emerging from running gradient descent for minimizing the population loss $L$, as in (Bietti et al., 2023; Chen et al., 2025). This raises a natural question: if we use the reparameterization on $\boldsymbol{U}$ and $\boldsymbol{V}$ but not $\boldsymbol{W}$, what is the implicit bias of running gradient descent on $\boldsymbol{W}$? In Theorem 3.4 (full proof in Appendix D.4), we show that gradient desecent does *not* directionally converges to $\sum_q E(q)\tilde{E}^\top(q)$.

**Theorem 3.4.** *For any $N \geq 4$, there exists a problem instance in the noiseless setting such that with $\mathcal{Q} = \{q\}, |\mathcal{O}| = 2, \boldsymbol{W}^* := E(q)\tilde{E}^\top(q), \boldsymbol{U}$ and $\boldsymbol{V}$ defined in Lemma 3.1, running gradient descent on $\boldsymbol{W}$ from $\boldsymbol{W}_0 = \mathbf{0}$ satisfies $\lim_{t \to \infty}\left|\langle\frac{\boldsymbol{W}_t}{\|\boldsymbol{W}_t\|}, \frac{\boldsymbol{W}^*}{\|\boldsymbol{W}^*\|}\rangle\right| = \frac{2}{\sqrt{12}} < 1.$*

## 4 Noiseless Learning with Softmax Attention

Fix a sentence with trigger word $q$ and output word $y$. Recall that in order to achieve a zero population logistic loss, it is necessary that the logits $\xi_y$ of the output word $y$ approaches infinity. The proof

sketch of Lemma 3.1 in the previous section showed that having an *unbounded* attention scores $\sigma(\boldsymbol{x}_H^\top \boldsymbol{W} \boldsymbol{x}_h)$ at $z_{h-1} = q$ naturally allows $\xi_y$ to approach infinity. While this unboundedness hold for linear and ReLU attention, it does not hold for softmax attention, where the attention scores are bounded in the range $[0, 1]$. The following lemma (proof in Appendix E) shows a modified parameterization that can drive the logits $\xi_y$ to infinity under softmax attention.

**Lemma 4.1.** *Let* $s \in \mathbb{R}$ *and* $\boldsymbol{\lambda} = \{\lambda_k \in \mathbb{R}_+ : k \in \mathcal{Q}\}$. *By setting* $\boldsymbol{U} = [E(1) \, E(2) \, \ldots \, E(N)]^\top, \boldsymbol{V} = s\boldsymbol{I}_d$ *and* $\boldsymbol{W} = \sum_{k \in \mathcal{Q}} \lambda_k E(k) \left( \tilde{E}(k)^\top - \sum_{x=1, x \neq k}^N \tilde{E}(x)^\top \right)$, *for softmax attention, we obtain* $\lim_{s \to \infty} \lim_{(\lambda_k)_{k \in \mathcal{Q}} \to \infty} L(\boldsymbol{V}, \boldsymbol{W}) = 0$.

This new parameterization has two modifications compared to the one in Lemma 3.1: a subtraction $-\sum_{x=1, x \neq k}^N \tilde{E}(x)^\top$ from $\boldsymbol{W}$, and a new scaling factor $s \in \mathbb{R}$ in the value matrix $\boldsymbol{V}$. The first modification ensures that $\boldsymbol{x}_H^\top \boldsymbol{W} \boldsymbol{x}_h$ tends to positive and negative infinity for positions $h$ where $z_{h-1} = q$ and $z_{h-1} \neq q$, respectively. Correspondingly, the attention scores $\sigma(\boldsymbol{x}_H^\top \boldsymbol{W} \boldsymbol{x}_h)$ tends to 1 and 0 for these two cases. The second modification effectively shifts the scaling in $\xi$ from $\boldsymbol{W}$ to $\boldsymbol{V}$, and implies that the desired optimality comes from $\lim_{s \to \infty} s \cdot 1 = \infty$ and $\lim_{s \to \infty} s \cdot 0 = 0$.

Note that in the statement of Lemma 4.1, the order of the two limit operations are strict and not exchangeable. Therefore, it is an important question that whether this optimality can actually be realized by running normalized gradient descent on $L(\boldsymbol{V}, \boldsymbol{W})$. The following result answers this question in the positive, showing that from a certain initialization, both $s$ and $(\lambda_k)_k$ approaches infinity. Moreover, the population loss $L(\boldsymbol{V}, \boldsymbol{W})$ converges to 0 at a linear rate.

**Theorem 4.2.** *Let* $\eta > 0$ *and* $T_0 = \lceil \frac{|\mathcal{Q}| \ln H}{2\eta} \rceil$. *By running* $t$ *rounds normalized gradient descent with learning rate* $\eta$ *from initialization* $\lambda_{q,0} = 0$ *for all* $q \in \mathcal{Q}$ *and* $s_0 = \frac{|\mathcal{Q}| \ln H + 2}{2}$, *we obtain* $s_t \geq \frac{1}{2} + \eta(t - T_0)$ *and* $\lambda_{q,t} \geq \frac{\eta t}{|\mathcal{Q}|}$ *for any* $t > T_0$. *Moreover, this implies* $L(\boldsymbol{V}, \boldsymbol{W}) \leq O(N \exp(-\eta t / |\mathcal{Q}|))$.

The proof of Theorem 4.2 divides the training process into two distinct phases: $t \leq T_0$ and $t > T_0$. During this first phase, the signs of the derivatives $\frac{\partial L}{\partial s_t}$ may oscillate between $+1$ and $-1$ while $\lambda_{q,t}$ and the attention scores grow quickly. In the second phase, $\lambda_{q,t}$ has become sufficiently large so that the attention scores become more stable, allowing $s_t$ to increase monotonically. Similar stage-wise convergence analysis of transformers with softmax attention has been observed before in other tasks such as regression (Huang et al., 2024a) and binary classification (Huang et al., 2025b).

## 5 NOISY LEARNING

Recall that our noisy data model, where $\alpha > 0$, is a variant of the setting in Chen et al. (2025). As an upgrade from Chen et al. (2025), we allow *multiple* trigger words, i.e. $|\mathcal{Q}| > 1$. Moreover, we allow *any* distributions of bigrams $(q, y)$ and $(q, \tau)$ in the sentence, and do not assume that the ratio of the frequencies of the bigrams $(q, \tau)$ and $(q, y)$ is $\frac{\alpha}{1-\alpha}$. In other words, our results are based on the distribution of the label $z_{H+1}$ instead of the distribution of the bigrams $(q, \tau)$ and $(q, y)$. We emphasize that a Bayes-optimal solution for our noisy data model will also be a Bayes-optimal solution for the model in Chen et al. (2025).

### 5.1 APPROXIMATION CAPABILITIES OF ONE-LAYER TRANSFORMERS ON NOISY SETTING

First, we show that by relying on just the distribution of $z_{H+1}$, the model defined in (1) is capable of making the population loss arbitrarily close to the loss of the Bayes optimal strategy. Fix a trigger word $q$ and an output word $y$. The label of a sentence that contains both $(q, \tau)$ and $(q, y)$ is either $\tau$ with probability $\alpha$ or $y$ with probability $1 - \alpha$, independent of other tokens in the sentence. Thus, the Bayes optimal strategy is to predict $\tau$ and $y$ with the same probabilities. Let $\hat{y}$ be the prediction of this strategy. Its expected loss is equal to the entropy $L_{\text{Bayes}} = -\alpha \ln \alpha - (1 - \alpha) \ln(1 - \alpha)$. Similar to the previous sections, we set and fix the unembedding layer $\boldsymbol{U} = [E(1) \, E(2) \, \ldots \, E(N) \, E(N+1)]^\top$. The following two lemmas demonstrate that the Bayes optimality of specific parameterization for linear/ReLU and softmax attentions.

**Lemma 5.1.** *Let* $\lambda, \gamma \in \mathbb{R}$. *By setting* $\boldsymbol{V} = \boldsymbol{I}_d, \boldsymbol{W} = \lambda \sum_{q \in \mathcal{Q}} E(q)(\tilde{E}^\top(q) - E^\top(\tau))$ *and* $\boldsymbol{F} = E(\tau)(\sum_{q \in \mathcal{Q}} \gamma E^\top(q) + \tilde{E}^\top(q))$ *and using linear or ReLU attention, we obtain*

$$\lim_{\lambda \to \infty, \gamma \to \ln \frac{\alpha}{1-\alpha}} L(\lambda, \gamma) := \lim_{\lambda \to \infty, \gamma \to \ln \frac{\alpha}{1-\alpha}} \mathbb{E}_{y, z_{1:H+1}} \left[ -\ln \frac{\exp(\xi_{z_{H+1}})}{\sum_{j \in [N+1]} \exp(\xi_j)} \right] = L_{\text{Bayes}}.$$

**Lemma 5.2.** *Let $s, \lambda, \gamma \in \mathbb{R}$. By setting $\boldsymbol{V} = s\boldsymbol{I}_d, \boldsymbol{W} = \lambda \sum_{q \in \mathcal{Q}} E(q)(\tilde{E}^\top(q) - 2E^\top(\tau) - \sum_{x=1, x \neq q}^N \tilde{E}(x)^\top)$ and $\boldsymbol{F} = E(\tau)(\sum_{q \in \mathcal{Q}} \gamma E^\top(q) + \tilde{E}^\top(q))$ and using softmax attention,*

$$\lim_{s \to \infty, \lambda \to \infty, \gamma \to \ln \frac{\alpha}{1-\alpha}} L(s, \lambda, \gamma) := \lim_{s \to \infty} \lim_{\lambda \to \infty, \gamma \to \ln \frac{\alpha}{1-\alpha}} \mathbb{E}_{y, z_{1:H+1}} \left[ -\ln \frac{\exp(\xi_{z_{H+1}})}{\sum_{j \in [N+1]} \exp(\xi_j)} \right] = L_{\text{Bayes}}.$$

*Remark* 5.3. These results are distribution-agnostic in the sense that they hold for any word distribution as long as the conditions of the data model are satisfied. Moreover, they hold even for $\alpha > 0.5$. This is a major advantage over the existing results in Chen et al. (2025), which required $\alpha \leq 0.5$. Setting $\alpha > 0.5$ also reflects a wider range of practical scenarios where the generic bigrams such as "to the" often appear more frequently than context-dependent bigrams such as "to Bob".

### 5.2 TRAINING DYNAMICS AND ON-CONVERGENCE BEHAVIOR: A FINITE-SAMPLE ANALYSIS

For ease of exposition, we focus on linear and ReLU attentions. The analysis for softmax attention follows a nearly identical proof with an additional beginning phase similar to the analysis in Section 4. Let $M$ denote the size of a dataset of i.i.d. sentences $z_{H+1}$ generated from the data model in Definition 2.1. Instead of minimizing the full population loss as in existing works (Bietti et al., 2023; Huang et al., 2024a; Chen et al., 2025), which would require either knowing $\alpha$ or taking $M \to \infty$, we aim to derive a finite-sample analysis that holds for finite $M$ and unknown $\alpha$.

Before presenting our training algorithm and its finite-sample analysis, we first discuss the easier case where $\alpha$ is known and explain why it is difficult to derive the convergence rate of the population loss in (67). Observe that the function $f_C(\lambda, \gamma) = -C\lambda - \alpha\gamma + \ln(e^{C\lambda} + e^{C\lambda+\gamma} + N - 1)$ is jointly convex and 1/2-smooth with respect to $\lambda$ and $\gamma$. Hence, at first glance, it seems that the convergence rate of $L(\lambda, \gamma) = \mathbb{E}_C[f_C(\lambda, \gamma)]$ follows from existing results in (stochastic) convex and smooth optimization (Nemirovski et al., 2009). However, as a convex function on unbounded domain with negative partial derivatives, no *finite* minimizer exists for $L(\lambda, \gamma)$. This implies that minimizing $L(\lambda, \gamma)$ is a multi-dimensional convex optimization problem on astral space (Dudík et al., 2022). To our knowledge, nothing is known about the convergence rate to the *infimum* for this problem.

**Training Algorithm.** Our approach for solving this astral space issue is to estimate $\gamma$ directly from the dataset and then run normalized gradient descent on $\lambda$. More specifically, recall that $M$ is the number of i.i.d. sentences in the training set. We use the superscript $(m)$ to denote quantities that belong to the $m$-th sentence, where $m \in [M]$. Let $M_\tau = \sum_{m=1}^M \mathbb{1}\{z_{H+1}^{(m)} = \tau\}$ be the number of sentences where $z_{H+1}$ is $\tau$. Let $\hat{\alpha} = \frac{M_\tau}{M}$ and $\hat{\gamma} = \ln \frac{\hat{\alpha}}{1-\hat{\alpha}}$ be the unbiased estimates for $\alpha$ and $\gamma = \ln \frac{\alpha}{1-\alpha}$, respectively. We use $\hat{\gamma}$ in the parameterization of $\boldsymbol{F}$, i.e. $\boldsymbol{F} = E(\tau)(\sum_{q \in \mathcal{Q}} \hat{\gamma} E^\top(q) + \tilde{E}^\top(q))$.

The empirical loss is $L_{\text{emp}}(\lambda) = \frac{1}{M} \sum_{m=1}^M -\ln \frac{\exp(\xi_{z_{H+1}}^{(m)})}{\sum_{j \in [N+1]} \exp(\xi_j^{(m)})}$. Then, we run normalized gradient descent on $L_{\text{emp}}(\lambda)$ with a constant learning rate $\eta > 0$. The formal procedure is given in Algorithm 1 in Appendix F.2. The following theorem shows that the population loss converges at a linear rate to the Bayes risk, similar to the noiseless setting.

**Theorem 5.4.** *With probability at least $1 - \delta$ over the training set of size $M$, after $t$ iterations of normalized gradient descent on $L_{\text{emp}}(\lambda)$, Algorithm 1 guarantees that*

$$L(s = 1, \lambda_t, \hat{\gamma}) \leq L_{\text{Bayes}} + \frac{1}{\min(\alpha, 1-\alpha) - \sqrt{\ln(2/\delta)/2M}} \frac{\ln(2/\delta)}{2M} + (N-1)e^{-\eta t}.$$

**OOD Generalization to unseen output words.** Similar to Theorem 3.3 for noiseless learning, the following theorem shows that Algorithm 1 produces a trained model that generalizes to an unseen output word $y_{\text{test}} \notin \mathcal{O} \cup \mathcal{Q}$ in the noisy setting. The proof can be found in Appendix F.3.

**Theorem 5.5.** *Fix any $y_{\text{test}} \in [N] \setminus (\mathcal{O} \cup \mathcal{Q})$. Take any **test** sentence generated by the noisy data model, except that every bigram $(q, y)$ is replaced with $(q, y_{\text{test}})$. Then, with probability at least $1 - \delta$, after $t$ iterations, Algorithm 1 returns a model that predicts $y_{\text{test}}$ and $\tau$ with probabilities*

$$\Pr[z_{H+1} = y_{\text{test}} \mid \lambda_t, \hat{\gamma}] := \frac{\exp(\xi_{y_{\text{test}}})}{\sum_{j \in [N+1]} \exp(\xi_j)} = 1 - \alpha + O\left(\sqrt{\frac{\ln(1/\delta)}{M}} + N^2 e^{-2\eta t}\right),$$

$$\Pr[z_{H+1} = \tau \mid \lambda_t, \hat{\gamma}] := \frac{\exp(\xi_\tau)}{\sum_{j \in [N+1]} \exp(\xi_j)} = \alpha + O\left(\sqrt{\frac{\ln(1/\delta)}{M}} + N^2 e^{-2\eta t}\right).$$

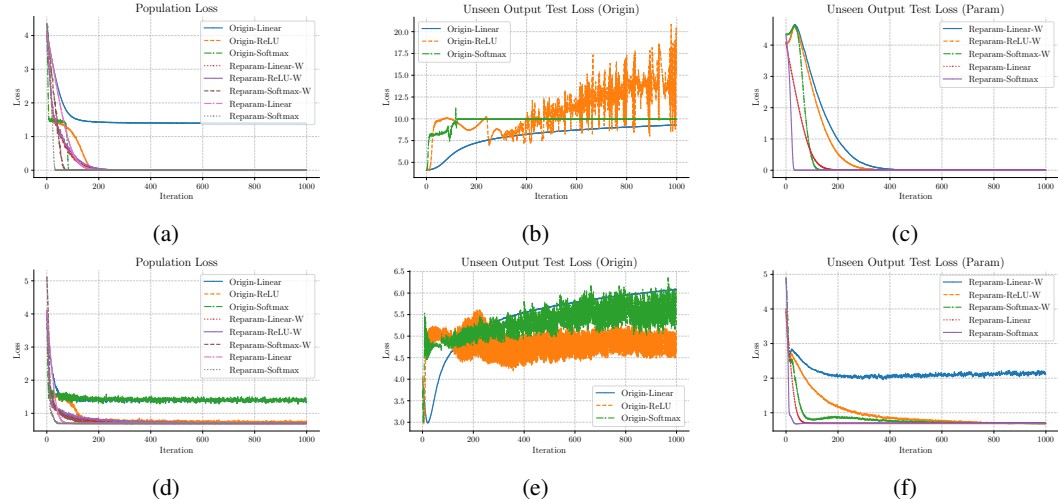

Figure 1: Population and OOD test loss of models trained on the population loss in noiseless and noisy settings. First row, left to right: the population loss, OOD test losses of `Origin` (no parameterization) models and OOD test losses of re-parameterized models in noiseless learning. Second row, left to right: the corresponding losses as in the first row in the noisy setting with $\alpha = 0.5$.

**Layer-wise Functionality.** An important empirical on-convergence behavior observed in Chen et al. (2025) is that after training, given a sentence which contains both bigrams $(q, y)$ and $(q, \tau)$, the feed-forward layer tends to predict the noise token $\tau$ while the attention layer tends to predict the output token $y$. This property can be expressed formally in terms of the final logits as

$$\xi_{A,y} > \max_{j \neq y} \xi_{A,j} \text{ and } \max_{j \neq \tau} \xi_{F,j} < \xi_{F,\tau}. \tag{4}$$

The following theorem shows that (4) always hold after a sufficiently large number of iterations.

**Theorem 5.6.** *Algorithm 1 guarantees that with probability $1 - \delta$, the condition (4) holds after any*
$$t \geq \max(1, \frac{1}{\eta} \left| \ln\left(1 - \alpha + \sqrt{\frac{\ln(2/\delta)}{2M}}\right) - \ln\left(\alpha - \sqrt{\frac{\ln(2/\delta)}{2M}}\right) \right|) \text{ training iterations.}$$

## 6 EMPIRICAL VALIDATION

| Models | Noiseless | | Noisy | |
|---|---|---|---|---|
| | 0-Loss? | Unseen $y$? | Bayes-Loss? | Unseen $y$? |
| Origin-Linear | — | — | — | — |
| Origin-ReLU | ✓ | — | — | — |
| Origin-Softmax | ✓ | — | — | — |
| Reparam-Linear-W | ✓ | ✓ | — | — |
| Reparam-ReLU-W | ✓ | ✓ | ✓ | ✓ |
| Reparam-Softmax-W | ✓ | ✓ | ✓ | ✓ |
| Reparam-Softmax | ✓ | ✓ | ✓ | ✓ |
| Reparam-Linear | ✓ | ✓ | ✓ | ✓ |
| Reparam-ReLU | ✓ | ✓ | ✓ | ✓ |

Table 1: Performance comparison of the original (no reparameterization) and parameterized models trained on the population loss. `0-Loss` and `Bayes-loss` indicate whether a model's Bayes-optimal in noiseless and noisy settings, respectively. `Unseen` $y$ indicate whether a model generalizes to unseen output words. The best models are highlighted.

To understand the impacts on empirical performances of the reparameterization presented in Sections 3 and 5, we evaluate the one-layer transformers (1) with different choices of parameterization and attention activation functions on the following data model.

**Data Model's Parameters.** We set the vocabulary size $N = 60$, the embedding dimension $d = 128$ and the context length $H = 256$. We use $Q = 5$ trigger words and $O = 4$ output words. The orthogonal embeddings are the standard basis vectors in $\mathbb{R}^d$. For the noisy setting, we choose $\alpha \in \{0.2, 0.5, 0.8\}$ to cover all three cases of $\alpha < 0.5, \alpha = 0.5$ and $\alpha > 0.5$. Our sentences are generated by picking uniformly at random a position for the bigram $(q, y)$ and a position for the bigram $(q, \tau)$ (in the noisy setting). All other words in a sentence are chosen uniformly at random from the set $[N] \setminus (\mathcal{Q} \cup \mathcal{O})$. For training on the population loss, we run normalized gradient descent with batch size $512$ over $T = 2000$ steps. For the finite-sample analysis, we train the models for $100$ epochs on a fixed training set of $M = 2048$ samples. Further details and results are in Appendix H.

**Baselines.** We compare 9 different models which differ in one or more following aspects: model type (`Reparam` versus `Origin`), attention type (linear versus ReLU versus Softmax), and whether the joint query-key matrix $\boldsymbol{W}$ is trained in full without reparameterization. More specifically, the term `Origin` refers to a model where the three trainable matrices $\boldsymbol{V}, \boldsymbol{W}$ and $\boldsymbol{F}$ are trained without reparameterization, while `Reparam` indicates that they are re-parameterized as in Lemmas 3.1, 4.1 and 5.1. Note that `Origin-Softmax` corresponds to the one-layer transformer in Chen et al. (2025). Finally, the model whose name contains both `Reparam` and `-W` has $\boldsymbol{V}$ and $\boldsymbol{F}$ re-parameterized but $\boldsymbol{W}$ is trained in full without any reparameterization.

### 6.1 Loss Convergence in Noiseless and Noisy Learning

We train 9 models, shown in Table 1, to minimize the population loss of noiseless and noisy settings. In the noiseless setting, only `Origin-Linear` fails to achieve a zero loss, indicating the limited approximation power of linear attention. In the noisy setting, 4 out of 9 models fail to achieve the Bayes risk. These four models consists of the three `Origin` models and `Reparam-Linear-W`. This shows that despite the high expressive power of one-layer transformers, gradient descent alone may not be able to find the *in-distribution* optimal solution. In contrast, all fully-parameterized one-layer models, including the one with linear attention, converge to Bayes-optimal solutions. This emphasizes the crucial role of structural parameterization in guiding gradient descent training towards better solutions, especially on models with low expressive power (i.e. linear attentions). In addition, Figures 1a and 1d show that for the models that converge to Bayes risk, their convergence rate is indeed linear. This empirically supports our theoretical claims in Theorems 3.2, 4.2 and 5.4.

### 6.2 OOD Generalization on Unseen Output Words

For each model, we examine whether their performance on seen output words in $\mathcal{O}$ is similar to that on unseen output words not in $\mathcal{O}$. Table 1 shows that the ability to generalize to unseen output words consistently increase with more parameterization and a model's expressiveness. On the other hand, for models with limited expressive power (i.e. one-layer linear transformers), parameterization plays a more important role. In particular, when all three matrices are trained without reparameterization, they collectively fail to generalize to unseen output words regardless of the type of attention. Figures 1b and 1e indicate that the test loss on unseen output words even *diverges* for original, non-reparameterized models. These results strongly suggest that (I) generalization to unseen output words is an important performance criterion for in-context recall learning, and (II) while one-layer transformers have the *representational capacity* to adapt to unseen output words, the solutions found by gradient descent are not naturally biased towards this adaptivity. On the other hand, Figures 1c and 1f shows that combining partial parameterization (on $\boldsymbol{V}$ and $\boldsymbol{F}$) and non-linear attention functions (e.g. ReLU or softmax) already leads to models that are simultaneously Bayes-optimal and generalize out-of-distribution. Moreover, similar to Bayes-optimality, OOD generalization can also be obtained with linear attention by careful parameterization. These empirical results hold for both $\alpha \leq 0.5$ and $\alpha > 0.5$, which further confirms the generality of our Theorem 5.5.

## 7 Conclusion

We studied the approximation capabilities of transformer for a one-step in-context recall task. Via a novel reparameterization regime, we rigorously proved that one-layer transformers are capable of achieving Bayes-optimal performance when being trained either directly on the population loss or on a finite dataset. Moreover, the same reparameterization allows one-layer transformers to generalize to sentences that are never seen during training. At the same time, our empirical results also show that without appropriate reparameterization, running gradient descent alone is unlikely to achieve non-trivial out-of-distribution generalization ability. Future works include an in-depth study on the theoretical guarantees and empirical performance of non-parameterized transformers that can simultaneously achieve Bayes-optimality and out-of-distribution generalization.

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

## A    ADDITIONAL EXAMPLE SENTENCES FOR THE IN-CONTEXT RECALL TASK

The in-context recall task considered in our paper encompasses a large range of practical linguistic scenarios. In this section, we provide additional example sentences in two domains: object identification and transitive inference. In all of these examples, each sentence contains at least one bigram $(q, y)$ before the last query word.

### A.1    OBJECT IDENTIFICATION

The task is to identify the right in-context object. Examples include:

Input: "To Harry" were the first two words in a letter that Ron and Hermione wrote to [?]
Output: Harry.

Input: People living the province of Quebec are proud of the natural beauty of the [?]
Output: province.

Input: You should travel on Sunday instead of on Monday, since there is a lot of traffic on [?]
Output: Monday.

### A.2    TRANSITIVE INFERENCE

The task is to identify the right object that has a specific relationship with other objects in the sentence. Examples include:

Input: If London is on the same continent as Paris, and Paris is on the same continent as Milan, then London is on the same continent as [?]
Output: Milan.

Input: If the table has the same color as the book, and the book has a different color than the chair, then the table has a different color than the [?]
Output: chair.

Input: If the GDP of Germany is larger than the combined GDP of Singapore and Spain, then it is certain that the GDP of Spain is smaller than the GDP of [?]
Output: Germany.

## B    COMPARISON TO THE LINEAR CONVERGENCE RATE RESULTS IN HUANG ET AL. (2024B)

In this section, we highlight the fundamental differences between our convergence rate results and those of Huang et al. (2024b), who also frame their convergence analysis in a next-token prediction problem and prove a linear convergence rate of the loss function (e.g. their Proposition 1).

- One-stage (ours) versus two-stage (theirs) training procedure : in our work, we train the (parameterized) key-query and value matrices simultaneously, which can also be seen as a one-stage training. In contrast, Huang et al. (their Algorithm 1) follow a two-stage training first train the value matrix for $T$ rounds, only then they train the key-query matrix.
- The absence (ours) or presence (theirs) of a hard-margin sub-problem: a key mechanism leading to the linear convergence rate in Huang et al. is the presence of a hard-margin sub-problem (see their Equation 2). This sub-problem arises out of their assumption that there exists a collocation (i.e., a one-to-one mapping) in their training sample. This assumption, and hence the hard-margin sub-problem, does not exist in our work.

Note that the two differences above are with respect to the *technical mechanism* in proving linear convergence rates. There are also *fundamental differences* in the problem setups between our work and Huang et al. (2024b), even though both belong to the category of next-token prediction tasks. In particular, our setup is an in-context noisy task where there could be multiple outputs for each sentence, while Huang et al. (2024b) studies a collocation-learning task where each token is always followed by an exact other token. In other words, Huang et al. (2024b) assumes the existence of an injective map between tokens, which is not the case in our setup.

## C    COMPARISON TO THE ANALYSIS IN BIETTI ET AL. (2023)

The nature of the theoretical results in Bietti et al. (2023) is fundamentally different from ours. In particular,

- The central theoretical results in Bietti et al. (2023) are their Lemma 1, Lemma 2 and Theorem 3. Their Lemmas 1 and 2 consider a very simple linear model with convex objective that does not involve any attention mechanism. Their Theorem 3 holds for sequential one-step gradient update on the population loss for the output, key and value matrices in that order. This sequential GD training on the matrices seem unnatural, and different from the simultaneous training procedure, albeit on re-parameterized matrices, in our work.
- Our task considers noisy output problems, where the output may be a noise token instead of a proper output. Bietti et al. (2023) consider noiseless problems only.
- Our task tests the model on unseen, out-of-distribution samples. Bietti et al. (2023) does not study out-of-distribution samples.
- Our analysis is both distribution-agnostic (for training on population samples) and finite-sample robust (for training on a finite dataset). Bietti et al. (2023)'s analysis uses specific, explicitly defined distributions (see their page 16, the first paragraph in appendix B.3), and does not consider finite-sample analysis.

## D    MISSING PROOFS IN SECTION 3

### D.1    PROOF OF LEMMA 3.1

First, we prove the following lemma on the attention scores. We write $\{a = b = c\}$ for the event that $a, b,$ and $c$ are equal, i.e. $\{a = b \cap b = c\}$.

**Lemma D.1.** *Under the reparameterization in Lemma 3.1, for all $h \in [H]$ a sentence with trigger word $q$, we obtain $\boldsymbol{x}_H^\top \boldsymbol{W} \boldsymbol{x}_h = \lambda_q \mathbb{1}\{z_{h-1} = z_H = q\}$.*

*Proof.* Let $\boldsymbol{W}_{|k} = \lambda_k E(k) \tilde{E}^\top(k)$. We have

$$\boldsymbol{W}_{|k} \boldsymbol{x}_h = \lambda_k E(k) \tilde{E}^\top(k)(E(z_h) + \tilde{E}(z_{h-1})) = \lambda_k E(k) \tilde{E}^\top(k) \tilde{E}(z_{h-1}) = \lambda_k E(k) \mathbb{1}\{z_{h-1} = k\}.$$

Hence, $\boldsymbol{x}_H^\top \boldsymbol{W}_{|k} \boldsymbol{x}_h = \lambda_k \mathbb{1}\{z_{h-1} = k\}(E(z_H) + \tilde{E}(z_{H-1}))^\top E(k) = \lambda_k \mathbb{1}\{z_{h-1} = z_H = k\}$. By construction, the sentence has only one trigger word $q$. We conclude that

$$\boldsymbol{x}_H^\top W \boldsymbol{x}_h = \sum_{k \in \mathcal{Q}} \boldsymbol{x}_H^\top W_{|k} \boldsymbol{x}_h = \lambda_q \mathbb{1}\{z_{h-1} = z_H = q\}.$$

$\square$

Lemma D.1 indicates that the attention scores are always non-negative. As a result, for both linear and ReLU attention, we have $\sigma(\boldsymbol{x}_H^\top W \boldsymbol{x}_h) = \boldsymbol{x}_H^\top W \boldsymbol{x}_h$. Hence, it suffices to prove Lemma 3.1 and our subsequent results for linear attention. More generally, our proof can be extended to any activation function where $\sigma(x) = cx$ for $c > 0, x \geq 0$.

*Proof.* (Of Lemma 3.1) Fix a trigger token $q \in \mathcal{Q}$ and an output token $y \in \mathcal{O}$. Consider sentences that contain $q$ and $y$ as their trigger and output, respectively. By Lemma D.1, for the linear attention model, we have

$$\xi_{A,j} = \boldsymbol{e}_j^\top \boldsymbol{U} \boldsymbol{V} \sum_{h=1}^{H} (\boldsymbol{x}_H^\top \boldsymbol{W} \boldsymbol{x}_h) \boldsymbol{x}_h = \boldsymbol{e}_j^\top \sum_{h=1}^{H} \lambda_q \mathbb{1}\{z_{h-1} = q\} \boldsymbol{U} \boldsymbol{x}_h = \boldsymbol{e}_j^\top \sum_{h=1}^{H} \lambda_q \mathbb{1}\{z_{h-1} = q\} \boldsymbol{e}_{z_h}.$$

Recall that $C_{q,y} = \sum_{h=1}^{H} \mathbb{1}\{z_{h-1} = q, z_h = y\} \geq 1$. We have $\mathbb{1}\{z_{h-1} = q\} \boldsymbol{e}_{z_h} = \mathbb{1}\{z_{h-1} = q\} \boldsymbol{e}_q$ because no tokens other than $y$ follows $q$ in each sentence by construction. Combining this with $\boldsymbol{e}_j^\top \boldsymbol{e}_q = \mathbb{1}\{j = q\}$, we obtain

$$\xi_{A,j} = \lambda_q C_{q,y} \mathbb{1}\{j = y\}. \tag{5}$$

Since $\xi_{F,j} = 0$ for $\boldsymbol{F} = \boldsymbol{0}$, we have $\xi_j = \xi_{A,j} + \xi_{F,j} = \xi_{A,j}$. This implies that the probability of predicting $y$ is $\lim_{\lambda_q \to \infty} \frac{\exp(\xi_y)}{\sum_{j \in [N]} \exp(\xi_j)} = \lim_{\lambda_q \to \infty} \frac{\exp(C_{q,y}\lambda_q)}{\exp(C_{q,y}\lambda_q) + N - 1} = 1$. □

### D.2 PROOF OF THEOREM 3.2

*Proof.* With linear attention, the population loss is defined as

$$
\begin{aligned}
L(\boldsymbol{\lambda}) &= \mathbb{E}_{q,y,z}\left[ -\ln \frac{\exp(C_{q,y}\lambda_q)}{\exp(C_{q,y}\lambda_q) + N - 1} \right] \\
&= \frac{1}{|\mathcal{Q}|} \sum_{q \in \mathcal{Q}} \mathbb{E}_{y,z}\left[ -\ln \frac{\exp(C_{q,y}\lambda_q)}{\exp(C_{q,y}\lambda_q) + N - 1} \right] \\
&= \frac{1}{|\mathcal{Q}|} \sum_{q \in \mathcal{Q}} \mathbb{E}_{y,z}[\ln(\exp(C_{q,y}\lambda_q) + N - 1) - C_{q,y}\lambda_q].
\end{aligned}
\tag{6}
$$

For each $q \in \mathcal{Q}$, the partial derivative of $L$ with respect to $\lambda_q$ is

$$
\frac{\partial L}{\partial \lambda_q} = \mathbb{E}_{y,z}\left[ C_{q,y}\left( \frac{\exp(C_{q,y}\lambda_q)}{\exp(C_{q,y}\lambda_q) + N - 1} - 1 \right) \right].
\tag{7}
$$

It follows that the normalized gradient descent update is

$$
\boldsymbol{\lambda}_{t+1} = \boldsymbol{\lambda}_t - \eta \frac{\nabla_{\boldsymbol{\lambda}} L}{\|\nabla_{\boldsymbol{\lambda}} L\|_2}
\tag{8}
$$

where $t = 0, 1, 2 \ldots$ denote the number of iterations, $\eta$ is a constant learning rate and $\nabla_{\boldsymbol{\lambda}} L = [\frac{\partial L}{\partial \lambda_1} \cdots \frac{\partial L}{\partial \lambda_{|\mathcal{Q}|}}]^\top$. We intialize $\boldsymbol{\lambda}_0 = \boldsymbol{0}$.

From Equation (7), we obtain that the partial derivatives are always negative and thus all $(\lambda_q)_q$ increases monotonically from 0. Next, we will show that $\lambda_{q,t} = \Omega(t)$ for all $q \in \mathcal{Q}, t \geq 0$. Initially, at $t = 0$ we have $\lambda_{1,t} = \lambda_{2,t} = \cdots = \lambda_{|\mathcal{Q}|,t}$. Assume that this property holds for some $t \geq 0$, for any $1 < k \leq |\mathcal{Q}|$, we have

$$
\begin{aligned}
\frac{\partial L}{\partial \lambda_{1,t}} &= \mathbb{E}_{y,z}\left[ C_{q_1,y} \frac{\exp(C_{q_1,y}\lambda_{1,t})}{\exp(C_{q_1,y}\lambda_{1,t}) + N - 1} - 1 \right] \\
&= \mathbb{E}_{y,z}\left[ C_{q_1,y} \frac{\exp(C_{q_1,y}\lambda_{k,t})}{\exp(C_{q_1,y}\lambda_{k,t}) + N - 1} - 1 \right] \\
&= \mathbb{E}_{y,z}\left[ C_{q_k,y} \frac{\exp(C_{q_k,y}\lambda_{k,t})}{\exp(C_{q_k,y}\lambda_{k,t}) + N - 1} - 1 \right] \\
&= \frac{\partial L}{\partial \lambda_{k,t}},
\end{aligned}
$$

where the third equality is from the symmetry in the distribution of the triggers. This implies that $\lambda_{1,t+1} = \lambda_{2,t+1} = \cdots = \lambda_{|\mathcal{Q}|,t+1}$. As a result, for each $q$,

$$
\frac{1}{\|\nabla_{\boldsymbol{\lambda}} L\|_2} \frac{\partial L}{\partial \lambda_{q,t}} = \frac{1}{\sqrt{|\mathcal{Q}|(\frac{\partial L}{\partial \lambda_{q,t}})^2}} \frac{\partial L}{\partial \lambda_{q,t}} = \frac{-1}{\sqrt{|\mathcal{Q}|}}.
$$

Therefore, $\lambda_{q,t} = \sum_{s=0}^{t-1} \frac{\eta}{\sqrt{|\mathcal{Q}|}} = \frac{\eta t}{\sqrt{|\mathcal{Q}|}} = \Omega(\eta t/|Q|)$. Plugging this into (6), we obtain

$$
L(\boldsymbol{\lambda}_t) = O\left( \ln\left( 1 + \frac{N-1}{\exp(\eta t/|Q|)} \right) \right) = O(N \exp(-\eta t/|Q|)).
$$

□

### D.3 PROOF OF THEOREM 3.3

*Proof.* Since Lemma D.1 holds for any set of output tokens, replacing $y$ by $y_{\text{test}}$ everywhere does not affect the attention scores $\boldsymbol{x}_H \boldsymbol{W} \boldsymbol{x}_h = \lambda_q \mathbb{1}\{z_{h-1} = q\}$. This implies that by Equation 5, we have $\xi_j = \xi_{A,j} = \lambda_q C_{q,j} \mathbb{1}\{j = y_{\text{test}}\}$. Therefore,

$$
\begin{aligned}
\Pr[z_{H+1} = y_{\text{test}} \mid \boldsymbol{\lambda}_t] &:= \frac{\exp(C_{q,y_{\text{test}}} \lambda_{q,t})}{\exp(C_{q,y_{\text{test}}} \lambda_{q,t}) + N - 1} = \frac{\exp(C_{q,y_{\text{test}}} \lambda_{q,t})}{\exp(C_{q,y_{\text{test}}} \lambda_{q,t}) + N - 1} \\
&\geq \frac{\exp(\lambda_{q,t})}{\exp(\lambda_{q,t}) + N - 1} = \frac{\exp(\eta t / |Q|)}{\exp(\eta t / |Q|) + N - 1},
\end{aligned}
$$

where the inequality is from $C_{q,y_{\text{test}}} \geq 1$ and the function $f(C) = \frac{\exp(Cx)}{\exp(Cx) + N - 1}$ is increasing in $C$ for $x, N > 0$. The last equality is due to $\lambda_{q,t} = \eta t / |Q|$. □

### D.4 DIRECTIONAL CONVERGENCE OF RUNNING GRADIENT DESCENT ON THE JOINT QUERY-KEY MATRIX

First, we introduce a variant of the data model in Definition 2.1. The set of trigger words contain only one element e.g. $Q = \{q\}$. The set of output words contain two elements $\mathcal{O} = \{y_1, y_2\}$. In addition to the set of trigger tokens $Q$ and the set of output tokens $\mathcal{O}$, we define a non-empty set of *neutral* tokens $\mathcal{N}$ so that $\mathcal{N} \cap (Q \cup \mathcal{O}) = \emptyset$. Fix an element $\square \in \mathcal{N}$. The data model is as below:

- Sample an output word $y \sim \text{Unif}(\mathcal{O})$.
- Sample a position $\zeta \sim \text{Unif}([H-3])$ and set $z_\zeta = q, z_{\zeta+1} = y$.
- Sample $z_h \sim \text{Unif}(\mathcal{N})$ for $h \in [H-2] \setminus \{\zeta, \zeta+1\}$.
- Set $z_{H-1} = \square$ and $z_H = q$.

We remark that this variant is a special case of the data model in Section 3, where each sentence has exactly one $(q, y)$ bigram and contain no output tokens other than $y$ (given that $y$ are the sampled output words).

*Proof.* (Of Theorem 3.4) Let $t \geq 0$ be the index of an iteration where $W_t$ satisfies $\boldsymbol{x}_H^\top \boldsymbol{W}_t \boldsymbol{x}_h = 0$ whenever $z_{h-1} \neq q$. Obviously, this trivially holds at $t = 0$. We will show that this property hold for $W_{t+1}$, and thus it holds throughout the gradient descent optimization process.

Keeping the reparameterization of $\boldsymbol{U} = [E(1)\ E(2)\ \ldots\ E(N)]^\top, \boldsymbol{V} = \boldsymbol{I}_d$ and using $\boldsymbol{e}_j^\top \boldsymbol{U} \boldsymbol{x}_h = \mathbb{1}\{z_h = j\}$, we write the population loss $L_t := L(\boldsymbol{W}_t)$ as

$$
\begin{aligned}
L_t &= \mathbb{E}_{y,z}\left[ -\ln \frac{\exp\left(\boldsymbol{e}_y^\top \boldsymbol{U} \boldsymbol{V} \sum_{h=1}^H (\boldsymbol{x}_H^\top \boldsymbol{W}_t \boldsymbol{x}_h) \boldsymbol{x}_h\right)}{\sum_{j \in [N]} \exp\left(\boldsymbol{e}_j^\top \boldsymbol{U} \boldsymbol{V} \sum_{h=1}^H (\boldsymbol{x}_H^\top \boldsymbol{W}_t \boldsymbol{x}_h) \boldsymbol{x}_h\right)} \right] \\
&= \mathbb{E}_{y,z}\left[ -\boldsymbol{e}_y^\top \boldsymbol{U} \sum_{h=1}^H (\boldsymbol{x}_H^\top \boldsymbol{W}_t \boldsymbol{x}_h) \boldsymbol{x}_h + \ln\left(\sum_{j \in [N]} \exp\left(\boldsymbol{e}_j^\top \boldsymbol{U} \sum_{h=1}^H (\boldsymbol{x}_H^\top \boldsymbol{W}_t \boldsymbol{x}_h) \boldsymbol{x}_h\right)\right) \right] \\
&= \mathbb{E}_{y,z}\left[ -\sum_{h=1}^H (\boldsymbol{x}_H^\top \boldsymbol{W}_t \boldsymbol{x}_h) \mathbb{1}\{z_h = y\} + \ln\left(\sum_{j \in [N]} \exp\left(\sum_{h=1}^H (\boldsymbol{x}_H^\top \boldsymbol{W}_t \boldsymbol{x}_h) \mathbb{1}\{z_h = j\}\right)\right) \right] \\
&= \mathbb{E}_{y,\zeta}\left[ -(\boldsymbol{x}_H^\top \boldsymbol{W}_t \boldsymbol{x}_{\zeta+1}) + \ln\left(\exp(\boldsymbol{x}_H^\top \boldsymbol{W}_t \boldsymbol{x}_{\zeta+1}) + N - 1\right) \right],
\end{aligned}
$$

where the last equality is due to the fact that

- $\mathbb{1}\{z_h = y\} = 1$ for $h = \zeta + 1$, and $\mathbb{1}\{z_h = y\} = 0$ otherwise.
- If $j \neq y$ then $z_{h-1} \neq q$. By the induction assumption, this implies $(\boldsymbol{x}_H^\top \boldsymbol{W}_t \boldsymbol{x}_h) \mathbb{1}\{z_h = j\} = 0$ for all $j \neq y$.

Taking the differential on both sides, we obtain

$$dL_t = \mathbb{E}_{y,\zeta} \left[ -(\boldsymbol{x}_H^\top(d\boldsymbol{W}_t)\boldsymbol{x}_{\zeta+1}) + \frac{\exp\big(\boldsymbol{x}_H^\top\boldsymbol{W}_t\boldsymbol{x}_{\zeta+1}\big)(\boldsymbol{x}_H^\top(d\boldsymbol{W}_t)\boldsymbol{x}_{\zeta+1})}{\exp\big(\boldsymbol{x}_H^\top\boldsymbol{W}_t\boldsymbol{x}_{\zeta+1}\big) + N - 1} \right]$$

$$= \mathbb{E}_{y,\zeta} \left[ -(\boldsymbol{x}_H^\top(d\boldsymbol{W}_t)\boldsymbol{x}_{\zeta+1}) + (\boldsymbol{x}_H^\top(d\boldsymbol{W}_t)\boldsymbol{x}_{\zeta+1})\hat{p}_y \right]$$

$$= \mathbb{E}_{y,\zeta} \left[ (\hat{p}_{y,t} - 1)(\boldsymbol{x}_H^\top(d\boldsymbol{W}_t)\boldsymbol{x}_{\zeta+1}) \right],$$

where $\hat{p}_{y,t} = \frac{\exp(\boldsymbol{x}_H^\top\boldsymbol{W}_t\boldsymbol{x}_{\zeta+1})}{\exp(\boldsymbol{x}_H^\top\boldsymbol{W}_t\boldsymbol{x}_{\zeta+1})+N-1}$ is the probability that the attention layer predicts $y$. As a result, the gradient of $L_t$ with respect to $\boldsymbol{W}_t$ is

$$\frac{dL_t}{d\boldsymbol{W}_t} = \mathbb{E}_{y,\zeta} \left[ (\hat{p}y, t - 1)(\boldsymbol{x}_H\boldsymbol{x}_{\zeta+1}^\top) \right]$$

$$= \mathbb{E}_{y,\zeta} \left[ (\hat{p}y, t - 1) \left( E(q) + \tilde{E}(\square)(E(y) + \tilde{E}(q))^\top \right) \right]$$

$$= \mathbb{E}_{y,\zeta} \left[ (\hat{p}y, t - 1) \left( E(q) + \tilde{E}(\square)(E^\top(y) + \tilde{E}^\top(q)) \right) \right]$$

$$= \frac{1}{2}\mathbb{E}_\zeta \left[ (\hat{p}_{y_1,t} - 1) \mid y = y_1 \right] \left( E(q) + \tilde{E}(\square)(E^\top(y_1) + \tilde{E}^\top(q)) \right)$$

$$+ \frac{1}{2}\mathbb{E}_\zeta \left[ (\hat{p}_{y_2,t} - 1) \mid y = y_2 \right] \left( E(q) + \tilde{E}(\square)(E^\top(y_2) + \tilde{E}^\top(q)) \right)$$

Due to the statistical symmetry between $y_1$ and $y_2$, we have $\mathbb{E}_\zeta \left[ (\hat{p}_{y_1,t} - 1) \mid y = y_1 \right] = \mathbb{E}_\zeta \left[ (\hat{p}_{y_2,t} - 1) \mid y = y_2 \right]$. Let $r_t = \mathbb{E}_\zeta \left[ (1 - \hat{p}_{y_1,t}) \mid y = y_1 \right]$. It follows that for some $r_t > 0$,

$$\frac{dL_t}{d\boldsymbol{W}_t} = r_t \sum_{y \in \{y_1, y_2\}} (E(q) + \tilde{E}(\square))(E^\top(y) + \tilde{E}^\top(q)). \tag{9}$$

Furthermore, running gradient descent

$$\boldsymbol{W}_{t+1} = \boldsymbol{W}_t - \eta \frac{dL_t}{d\boldsymbol{W}_t} \tag{10}$$

leads to $\boldsymbol{W}_{t+1} = \boldsymbol{W}_t + \eta r_t \sum_{y \in \{y_1, y_2\}} (E(q) + \tilde{E}(\square))(E^\top(y) + \tilde{E}^\top(q))$. In a sentence with output token $y$, for all $h \in [H]$ such that $z_{h-1} \neq q$, we have $z_h \neq y$. Hence,

$$(E^\top(y) + \tilde{E}^\top(q))\boldsymbol{x}_h = (E^\top(y) + \tilde{E}^\top(q))(E(z_h) + \tilde{E}(z_{h-1})) = 0. \tag{11}$$

As a result, for all $h$ where $z_{h-1} \neq q$, we have

$$\boldsymbol{x}_H^\top\boldsymbol{W}_{t+1}\boldsymbol{x}_h = \boldsymbol{x}_H^\top\boldsymbol{W}_t\boldsymbol{x}_h + \eta r_t\boldsymbol{x}_H^\top \sum_{y \in \{y_1, y_2\}} (E(q) + \tilde{E}(\square))(E^\top(y) + \tilde{E}^\top(q))\boldsymbol{x}_h \tag{12}$$

$$= 0. \tag{13}$$

By induction, we have Equation 9 holds for all $t$. Recall that we initialized $\boldsymbol{W}_0 = \boldsymbol{0}$. With a learning rate $\eta > 0$, running gradient descent results to

$$\boldsymbol{W}_t = (\sum_{s=0}^{t} r_t) \sum_{y \in \{y_1, y_2\}} (E(q) + \tilde{E}(\square))(E^\top(y) + \tilde{E}^\top(q)) \tag{14}$$

$$= R_t(E(q) + \tilde{E}(\square))(E^\top(y_1) + E^\top(y_2) + 2\tilde{E}^\top(q)) \tag{15}$$

$$= R_t\boldsymbol{A}, \tag{16}$$

where $R_t = \sum_{s=0}^{t} r_t \in \mathbb{R}_+$ is a positive number and $\boldsymbol{A} = (E(q) + \tilde{E}(\square))(E^\top(y_1) + E^\top(y_2) + 2\tilde{E}^\top(q))$. Thus, $\boldsymbol{W}_t$ is always in the same direction as $\boldsymbol{A}$. Hence,

$$\lim_{t \to \infty} \frac{\boldsymbol{W}_t}{\|\boldsymbol{W}_t\|} = \frac{\boldsymbol{A}}{\|\boldsymbol{A}\|}. \tag{17}$$

Next, recall that $\boldsymbol{W}^* = E(q)\tilde{E}^\top(q)$. Let $\boldsymbol{a}, \boldsymbol{b}, \boldsymbol{c}, \boldsymbol{d}, \boldsymbol{u} \in \mathbb{R}^d$ be five vectors corresponding to $E(q), E(y_1), E(y_2), \tilde{E}(q)$ and $\tilde{E}(\square)$, respectively. Note that these vectors are pairwise orthogonal unit vectors. The two matrices $\boldsymbol{A}$ and $\boldsymbol{W}^*$ are written as

$$\boldsymbol{A} = (a + u)(b^\top + c^\top + 2d^\top),$$
$$\boldsymbol{W}^* = ad^\top.$$

We will show that the Frobenius product $\langle \frac{\boldsymbol{A}}{\|\boldsymbol{A}\|}, \frac{\boldsymbol{W}^*}{\|\boldsymbol{W}^*\|} \rangle$ is not equal 1. We have

$$
\begin{aligned}
\langle \boldsymbol{A}, \boldsymbol{W}^* \rangle &= \mathrm{Tr}\big((W^*)^\top A\big) \\
&= \mathrm{Tr}\big(da^\top(a + u)(b^\top + c^\top + 2d^\top)\big) \\
&= \mathrm{Tr}\big(d(b^\top + c^\top + 2d^\top)\big) \\
&= \mathrm{Tr}\big((b^\top + c^\top + 2d^\top)d\big) \\
&= 2,
\end{aligned}
\tag{18}
$$

where the equalities follow from $a^T a = d^\top d = 1$ and the pairwise orthogonality. Furthermore,

$$
\begin{aligned}
\|\boldsymbol{A}\| &= \sqrt{\mathrm{Tr}(\boldsymbol{A}^\top \boldsymbol{A})} = \sqrt{\mathrm{Tr}((b + c + 2d)(a^\top + u^\top)(a + u)(b^\top + c^\top + 2d^\top))} \\
&= \sqrt{2\,\mathrm{Tr}((b + c + 2d)(b^\top + c^\top + 2d^\top))} \\
&= \sqrt{12},
\end{aligned}
$$

and

$$\|\boldsymbol{W}^*\| = \sqrt{\mathrm{Tr}((\boldsymbol{W}^*)^\top \boldsymbol{W}^*)} = \sqrt{\mathrm{Tr}(da^\top ad^\top)} = 1.$$

Obviously, $\langle \frac{\boldsymbol{A}}{\|\boldsymbol{A}\|}, \frac{\boldsymbol{W}^*}{\|\boldsymbol{W}^*\|} \rangle = \frac{2}{\sqrt{12}} < 1.$ $\qquad\square$

# E    MISSING PROOFS IN SECTION 4

## E.1    PROOF OF LEMMA 4.1

*Proof.* Consider a sentence with a trigger $q$ and output $y$. Similar to the proof of Lemma 3.1, we first compute the pre-softmax attention scores $\boldsymbol{x}_H^\top \boldsymbol{W} \boldsymbol{x}_h$. We have

$$
\boldsymbol{x}_H^\top \boldsymbol{W} \boldsymbol{x}_h = (E(q) + \tilde{E}(z_{h-1}))^\top \left( \sum_{q' \in \mathcal{Q}} \lambda_{q'} E(q') \left( \tilde{E}(q')^\top - \sum_{x=1, x \neq q'}^N \tilde{E}(x)^\top \right) \right) \boldsymbol{x}_h \tag{19}
$$

$$
= \lambda_q \left( \tilde{E}(q)^\top - \sum_{x=1, x \neq q}^N \tilde{E}(x)^\top \right) (E(z_h) + \tilde{E}(z_{h-1})) \tag{20}
$$

$$
= \lambda_q \left( \tilde{E}(q)^\top - \sum_{x=1, x \neq q}^N \tilde{E}(x)^\top \right) \tilde{E}(z_{h-1}) \tag{21}
$$

$$
= \lambda_q \left( \mathbb{1}\{z_{h-1} = q\} - \mathbb{1}\{z_{h-1} \neq q\} \right). \tag{22}
$$

It follows that

$$
\boldsymbol{x}_H^\top \boldsymbol{W} \boldsymbol{x}_h = \begin{cases} \lambda_q & \text{if } z_{h-1} = q, \\ -\lambda_q & \text{otherwise.} \end{cases} \tag{23}
$$

The attention score at the $h$-th token in a sentence is

$$
\sigma(\boldsymbol{x}_H^\top \boldsymbol{W} \boldsymbol{x}_h) = \frac{\exp\big(\boldsymbol{x}_H^\top \boldsymbol{W} \boldsymbol{x}_h\big)}{\sum_{j=1}^H \exp\big(\boldsymbol{x}_H^\top \boldsymbol{W} \boldsymbol{x}_j\big)} = \frac{\exp(\lambda_q(\mathbb{1}\{z_{h-1} = q\} - \mathbb{1}\{z_{h-1} \neq q\}))}{\sum_{j=1}^H \exp(\lambda_q(\mathbb{1}\{z_{h-1} = q\} - \mathbb{1}\{z_{h-1} \neq q\}))} \tag{24}
$$

$$
= \begin{cases} \frac{\exp(\lambda_q)}{C_{q,y}\exp(\lambda_q) + (H - C_{q,y})\exp(-\lambda_q)} & \text{if } z_{h-1} = q, \\ \frac{\exp(-\lambda_q)}{C_{q,y}\exp(\lambda_q) + (H - C_{q,y})\exp(-\lambda_q)} & \text{otherwise.} \end{cases} \tag{25}
$$

Obviously, $\lim_{\lambda_q \to \infty} \sigma(\boldsymbol{x}_H^\top \boldsymbol{W} \boldsymbol{x}_h) = 1$ if $z_{h-1} = q$ and $\lim_{\lambda_q \to \infty} \sigma(\boldsymbol{x}_H^\top \boldsymbol{W} \boldsymbol{x}_h) = 0$ otherwise.

Next, we compute $\xi_j$ for $j = y$ and $j \neq y$. Recall that $\boldsymbol{V} = s\boldsymbol{I}_d$ and $\boldsymbol{e}_y^\top \boldsymbol{U} \boldsymbol{x}_h = \mathbb{1}\{z_h = y\}$. With $j = y$, we have

$$\xi_y = \boldsymbol{e}_y^\top \boldsymbol{U} \boldsymbol{V} \sum_{h=1}^{H} \sigma(\boldsymbol{x}_H \boldsymbol{W} \boldsymbol{x}_h) \boldsymbol{x}_h = s \sum_{h=1}^{H} \sigma(\boldsymbol{x}_H \boldsymbol{W} \boldsymbol{x}_h) \mathbb{1}\{z_h = y\} \tag{26}$$

$$= s \left( \sum_{z_{h-1}=q} \sigma(\boldsymbol{x}_H \boldsymbol{W} \boldsymbol{x}_h) \mathbb{1}\{z_h = y\} + \sum_{z_{h-1} \neq q} \sigma(\boldsymbol{x}_H \boldsymbol{W} \boldsymbol{x}_h) \mathbb{1}\{z_h = y\} \right) \tag{27}$$

$$= s \frac{C_{q,y} \exp(\lambda_q) + (C_y - C_{q,y}) \exp(-\lambda_q)}{C_{q,y} \exp(\lambda_q) + (H - C_{q,y}) \exp(-\lambda_q)}. \tag{28}$$

With $j \neq y$, we have

$$\xi_j = \boldsymbol{e}_j^\top \boldsymbol{U} \boldsymbol{V} \sum_{h=1}^{H} \sigma(\boldsymbol{x}_H \boldsymbol{W} \boldsymbol{x}_h) \boldsymbol{x}_h = s \sum_{h=1}^{H} \sigma(\boldsymbol{x}_H \boldsymbol{W} \boldsymbol{x}_h) \mathbb{1}\{z_h = j\} \tag{29}$$

$$= s \sum_{h=1}^{H} \frac{\exp(-\lambda_q)}{C_{q,y} \exp(\lambda_q) + (H - C_{q,y}) \exp(-\lambda_q)} \mathbb{1}\{z_h = j\} \tag{30}$$

$$= s \frac{C_j \exp(-\lambda_q)}{C_{q,y} \exp(\lambda_q) + (H - C_{q,y}) \exp(-\lambda_q)}. \tag{31}$$

The desired statement follows from the fact that $1 \leq C_{q,y} < H, 0 \leq C_j < H$ and thus $\lim_{\lambda_q \to \infty} \frac{C_{q,y} \exp(\lambda_q) + (C_y - C_{q,y}) \exp(-\lambda_q)}{C_{q,y} \exp(\lambda_q) + (H - C_{q,y}) \exp(-\lambda_q)} = 1$ and $\lim_{\lambda_q \to \infty} \frac{C_j \exp(-\lambda_q)}{C_{q,y} \exp(\lambda_q) + (H - C_{q,y}) \exp(-\lambda_q)} = 0$ for $j \neq y$. Hence,

$$\lim_{s \to \infty} \lim_{\lambda_q \to \infty} \xi_y = \lim_{s \to \infty} s = \infty \tag{32}$$

$$\lim_{s \to \infty} \lim_{\lambda_q \to \infty} \xi_j = \lim_{s \to \infty} 0 = 0. \tag{33}$$

$\square$

### E.2 PROOF OF THEOREM 4.2

*Proof.* Consider a sample sentence with trigger word $q$ and output word $y$. We use short-hand notations $A = C_{q,y}, B = C_y, x = \lambda$. Note that $1 \leq A < B < H$. The loss incurred by this sample

is

$$f(s, x) = -\ln \frac{\exp(\xi_y)}{\exp(\xi_y) + \sum_{i \neq x} \exp(\xi_i)}$$

$$= -\xi_y + \ln \left( \exp(\xi_y) + \sum_{i \neq y} \exp(\xi_i) \right)$$

$$= -s \frac{C_{q,y} \exp(x) + (C_y - C_{q,y}) \exp(-x)}{C_{q,y} \exp(x) + (H - C_{q,y}) \exp(-x)}$$

$$+ \ln \left( \exp \left( s \frac{C_{q,y} \exp(x) + (C_y - C_{q,y}) \exp(-x)}{C_{q,y} \exp(x) + (H - C_{q,y}) \exp(-x)} \right) + \sum_{i \neq y} \exp \left( s \frac{C_i \exp(-x)}{C_{q,y} \exp(x) + (H - C_{q,y}) \exp(-x)} \right) \right)$$

$$= -s \frac{A \exp(x) + (B - A) \exp(-x)}{A \exp(x) + (H - A) \exp(-x)}$$

$$+ \ln \left( \exp \left( s \frac{A \exp(x) + (B - A) \exp(-x)}{A \exp(x) + (H - A) \exp(-x)} \right) + \sum_{i \neq y} \exp \left( s \frac{C_i \exp(-x)}{A \exp(x) + (H - A) \exp(-x)} \right) \right)$$

$$= -s \frac{Ae^{2x} + B - A}{Ae^{2x} + H - A} + \ln \left( \exp \left( s \frac{Ae^{2x} + B - A}{Ae^{2x} + H - A} \right) + \sum_{i \neq y} \exp \left( s \frac{C_i}{Ae^{2x} + H - A} \right) \right).$$

$$(34)$$

Let $g(x) = Ae^{2x} + H - A$ and $u(x) = Ae^{2x} + B - A$. We have

$$f(s, x) = -s \frac{u(x)}{g(x)} + \ln \left( \exp \left( s \frac{u(x)}{g(x)} \right) + \sum_{i \neq y} \exp \left( s \frac{C_i}{g(x)} \right) \right). \qquad (35)$$

Let $v(s, x) = \exp \left( s \frac{u(x)}{g(x)} \right) + \sum_{i \neq y} \exp \left( s \frac{C_i}{g(x)} \right)$. Note that $v(s, x) > 0$ for all $s, x \in \mathbb{R}$.

Next, we compute the partial derivatives of $f$ with respect to $x$ and $s$. We have

$$\frac{dg}{dx} = 2Ae^{2x} \qquad (36)$$

$$\frac{du}{dx} = 2Ae^{2x} \qquad (37)$$

$$\frac{d\frac{u}{g}}{dx} = \frac{u'(x)g(x) - u(x)g'(x)}{g(x)^2} \qquad (38)$$

$$= \frac{2Ae^{2x}(Ae^{2x} + H - A) - 2Ae^{2x}(Ae^{2x} + B - A)}{g(x)^2} \qquad (39)$$

$$= \frac{2Ae^{2x}(H - B)}{g(x)^2} \qquad (40)$$

$$\frac{d\frac{1}{g}}{dx} = -\frac{g'(x)}{g(x)^2} = \frac{-2Ae^{2x}}{g(x)^2}. \qquad (41)$$

Additionally,

$$\frac{\partial v}{\partial x} = s \frac{d\frac{u}{g}}{dx} \exp \left( s \frac{u(x)}{g(x)} \right) + \sum_{i \neq y} sC_i \frac{d\frac{1}{g}}{dx} \exp \left( s \frac{C_i}{g(x)} \right) \qquad (42)$$

$$= s \left( \frac{2Ae^{2x}(H - B)}{g(x)^2} \exp \left( s \frac{u(x)}{g(x)} \right) + \sum_{i \neq y} -2C_i \frac{Ae^{2x}}{g(x)^2} \exp \left( s \frac{C_i}{g(x)} \right) \right) \qquad (43)$$

$$= \frac{2Ase^{2x}}{g(x)^2} \left( (H - B) \exp \left( s \frac{u(x)}{g(x)} \right) - \sum_{i \neq y} C_i \exp \left( s \frac{C_i}{g(x)} \right) \right), \qquad (44)$$

and

$$\frac{\partial v}{\partial s} = \frac{u(x)}{g(x)} \exp\left(s\frac{u(x)}{g(x)}\right) + \sum_{i \neq y} \frac{C_i}{g(x)} \exp\left(s\frac{C_i}{g(x)}\right) \tag{45}$$

$$= \frac{1}{g(x)} \left( u(x) \exp\left(s\frac{u(x)}{g(x)}\right) + \sum_{i \neq y} C_i \exp\left(s\frac{C_i}{g(x)}\right) \right) \tag{46}$$

It follows that

$$\frac{\partial f}{\partial x} = -s\frac{d\frac{u}{g}}{dx} + \frac{\frac{\partial v}{\partial x}}{v(x)} \tag{47}$$

$$= -\frac{2Ase^{2x}(H - B)}{g(x)^2} + \frac{1}{v(x)}\frac{2Ase^{2x}}{g(x)^2}\left((H - B)\exp\left(s\frac{u(x)}{g(x)}\right) - \sum_{i \neq y} C_i \exp\left(s\frac{C_i}{g(x)}\right)\right) \tag{48}$$

$$= -\frac{2Ase^{2x}}{g(x)^2 v(x)}\left(v(x)(H - B) - (H - B)\exp\left(s\frac{u(x)}{g(x)}\right) + \sum_{i \neq y} C_i \exp\left(s\frac{C_i}{g(x)}\right)\right) \tag{49}$$

$$= -\frac{2Ase^{2x}}{g(x)^2 v(x)}\left(\sum_{i \neq y}(H - B + C_i)\exp\left(s\frac{C_i}{g(x)}\right)\right). \tag{50}$$

Since $A > 0, H - B + C_i > 0$ and $v(x) > 0$, we have $\frac{\partial f}{\partial x} < 0$ whenever $s > 0$.

Next, we have

$$\frac{\partial f}{\partial s} = -\frac{u(x)}{g(x)} + \frac{\frac{\partial v}{\partial s}}{v(x)} \tag{51}$$

$$= \frac{1}{g(x)v(x)}\left(-u(x)v(x) + u(x)\exp\left(s\frac{u(x)}{g(x)}\right) + \sum_{i \neq y} C_i \exp\left(s\frac{C_i}{g(x)}\right)\right) \tag{52}$$

$$= \frac{1}{g(x)v(x)}\left(\sum_{i \neq y}(C_i - u(x))\exp\left(s\frac{C_i}{g(x)}\right)\right) \tag{53}$$

$$= -\frac{1}{g(x)v(x)}\left(\sum_{i \neq y}(Ae^{2x} + B - A - C_i)\exp\left(s\frac{C_i}{g(x)}\right)\right). \tag{54}$$

Obviously, if $x \geq \frac{\ln H}{2}$, then $Ae^{2x} \geq AH \geq H > C_i$, which implies that $\frac{\partial f}{\partial s} < 0$.

**Phase Analysis.** With $\eta \in (0, 1)$ be the learning rate, define $T_0 = \lceil\frac{|\mathcal{Q}|\ln H}{2\eta}\rceil$. Recall that $s$ is intialized by $s_0 = \frac{|\mathcal{Q}|\ln H + 2}{2}$ and $\lambda_{q,0} = 0$ for all $q$. We divide the training process into two phases: the first phase is from round $t = 1$ to $t = T_0$, and the second phase is from $t = T_0 + 1$ onwards.

- In the first phase $1 \leq t \leq T_0$, we first show that $s_t$ may fluctuate but is always positive. Recall that for scalar value of $s$, the normalize gradient descent is equal to sign descent $s_t = s_{t-1} - \eta \operatorname{sign}\frac{\partial L}{\partial s}$. In the worst case, the signs of the partial derivatives are always positive. It follows that

$$s_t \geq s_0 - \eta T_0 \geq s_0 - \frac{|\mathcal{Q}|\ln H + 1}{2} \geq \frac{1}{2}. \tag{55}$$

Thus, $s_t > 0$ always holds in the first phase. This implies that $\frac{\partial f}{\partial \lambda_q} < 0$ in the first phase. Therefore, the update formula $\lambda_{q,t} = \frac{\eta t}{|\mathcal{Q}|}$ always holds, which implies

$$\lambda_{q,T_0} = \frac{\eta T_0}{|\mathcal{Q}|} \geq \frac{\ln H}{2}. \tag{56}$$

This implies that $\frac{\partial L}{\partial s} < 0$.

- In the second phase $t > T_0$, we now have that the signs of all partial derivatives are negative. Therefore,

$$s_t \geq \frac{1}{2} + \eta(t - T_0), \tag{57}$$

$$\lambda_{q,t} = \frac{\eta t}{|\mathcal{Q}|}. \tag{58}$$

Plugging these into (35), we obtain

$$f(s_t, \lambda_{q,t}) = \ln\left(1 + \sum_{i \neq y} \frac{\exp\left(s_t \frac{C_i}{g(\lambda_{q,t})}\right)}{\exp\left(s_t \frac{u(\lambda_{q,t})}{g(\lambda_{q,t})}\right)}\right) = \sum_{i \neq y} \frac{\exp\left(s_t \frac{C_i}{g(\lambda_{q,t})}\right)}{\exp\left(s_t \frac{u(\lambda_{q,t})}{g(\lambda_{q,t})}\right)} \tag{59}$$

$$\leq N \frac{\exp\left(s_t \frac{H}{g(\lambda_{q,t})}\right)}{\exp\left(s_t \frac{u(\lambda_{q,t})}{g(\lambda_{q,t})}\right)} \leq O\left(N \frac{\exp\left(s_t \frac{H}{g(\lambda_{q,t})}\right)}{\exp(2s_t)}\right) \tag{60}$$

$$\leq O(N \exp(-\eta t)), \tag{61}$$

where the second inequality is from $\frac{u(x)}{g(x)} \geq \frac{1}{2}$ for sufficiently large $x$, and the last inequality is from $2s_t = \Omega(\eta t)$ and

$$s_t \frac{H}{g(\lambda_{q,t})} \leq (s_0 + \eta T_0 + \eta t) \frac{H}{C_{q,y} e^{2\lambda_{q,t}} + H - C_{q,y}} \tag{62}$$

$$= (s_0 + \eta T_0 + \eta t) \frac{H}{C_{q,y} e^{2\eta t/|\mathcal{Q}|} + H - C_{q,y}} \tag{63}$$

$$\leq O(1). \tag{64}$$

$\square$

# F  MISSING PROOFS IN SECTION 5

## F.1  PROOF OF LEMMA 5.1

We prove the following two lemmas on the optimality of linear and softmax attention for the noisy setting.

**Lemma F.1.** *(Optimality of linear and ReLU attention for noisy task) Under the reparameterization regime defined in Lemma 5.1, for all $\alpha \in (0,1)$, using linear and ReLU attention, we obtain*

$$\lim_{\lambda \to \infty, \gamma \to \ln \frac{\alpha}{1-\alpha}} L(\lambda, \gamma) := \lim_{\lambda \to \infty, \gamma \to \ln \frac{\alpha}{1-\alpha}} \mathbb{E}_{y, z_{1:H+1}}\left[-\ln \frac{\exp(\xi_{z_{H+1}})}{\sum_{j \in [N+1]} \exp(\xi_j)}\right] = L_{\text{Bayes}}. \tag{65}$$

*Proof.* Fix a sentence with trigger $q$ and output $y$. It suffices to show that

$$\xi_j = \mathbb{1}\{j = y \lor j = \tau\}(\lambda C_{q,y} + \mathbb{1}\{j = \tau\}\gamma), \tag{66}$$

since this implies

$$L(\lambda, \gamma) = \mathbb{E}_{q,y,z_{1:H+1}}\left[-\ln \frac{\exp(\xi_{z_{H+1}})}{\sum_{j \in [N+1]} \exp(\xi_j)}\right] = \mathbb{E}_y\left[\mathbb{E}_{z_{1:H+1}}\left[-\ln \frac{\exp(\xi_{z_{H+1}})}{\sum_{j \in [N+1]} \exp(\xi_j)} \mid y\right]\right]$$

$$= \mathbb{E}_{q,y}\left[\mathbb{E}_{z_{1:H}}\left[(\alpha - 1)\left(\ln \frac{e^{C_{q,y}\lambda}}{e^{C_{q,y}\lambda} + e^{C_{q,y}\lambda + \gamma} + N - 1}\right) - \alpha\left(\ln \frac{e^{C_{q,y}\lambda + \gamma}}{e^{C_{q,y}\lambda} + e^{C_{q,y}\lambda + \gamma} + N - 1}\right)\right]\right]. \tag{67}$$

The desired statement follows from the facts that

$$\lim_{\lambda \to \infty} \frac{e^{C\lambda}}{e^{C\lambda} + e^{C\lambda + \ln \frac{\alpha}{1-\alpha}} + N - 1} = 1 - \alpha, \text{ and } \lim_{\lambda \to \infty} \frac{e^{C\lambda + \ln \frac{\alpha}{1-\alpha}}}{e^{C\lambda} + e^{C\lambda + \ln \frac{\alpha}{1-\alpha}} + N - 1} = \alpha$$

for any bounded $0 \leq C \leq H$. Note that we require $\alpha$ strictly larger than 0 so that we can use $\lim_{\lambda \to \infty} \ln \left( \frac{e^{C_{q,y}\lambda + \gamma}}{e^{C_{q,y}\lambda} + e^{C_{q,y}\lambda + \gamma} + N - 1} \right) = \ln \left( \lim_{\lambda \to \infty} \frac{e^{C_{q,y}\lambda + \gamma}}{e^{C_{q,y}\lambda} + e^{C_{q,y}\lambda + \gamma} + N - 1} \right) = \ln \alpha$.

We turn to proving (66). Similar to the proof of Lemma 3.1, we start by examining the attention scores $\boldsymbol{x}_H^\top \boldsymbol{W} \boldsymbol{x}_h$ for $h = 1, 2, \ldots, H$. First, the product $\boldsymbol{x}_H^\top \boldsymbol{W}$ is equal to

$$(E(q)^\top + \tilde{E}(z_{H-1})^\top)\lambda \left( \sum_{q' \in \mathcal{Q}} E(q') \left( \tilde{E}^\top(q') - E^\top(\tau) \right) \right) \tag{68}$$

$$= \lambda E(q)^\top \left( \sum_{q' \in \mathcal{Q}} E(q') \left( \tilde{E}^\top(q') - E^\top(\tau) \right) \right) \tag{69}$$

$$= \lambda(\tilde{E}(q)^\top - E(\tau)^\top). \tag{70}$$

Next, we consider two cases:

- For $z_h = \tau$, we have $z_{h-1} = q$, therefore
$$\boldsymbol{x}_H^\top \boldsymbol{W} \boldsymbol{x}_h = \lambda \left( \tilde{E}^\top(q) - E^\top(\tau) \right) (E(\tau) + \tilde{E}(q)) = \boldsymbol{0}. \tag{71}$$

- For $z_h \in [N+1] \setminus \{\tau\}$, we have
$$\boldsymbol{x}_H^\top \boldsymbol{W} \boldsymbol{x}_h = \lambda \left( \tilde{E}^\top(q) - E^\top(\tau) \right) (E(z_h) + \tilde{E}(z_{h-1})) = \lambda \mathbb{1}\{z_{h-1} = q\}. \tag{72}$$

It follows that the attention scores are (using $\boldsymbol{x}_H = E(q) + \tilde{E}(z_{H-1})$)
$$\boldsymbol{x}_H^\top \boldsymbol{W} \boldsymbol{x}_h = \lambda \mathbb{1}\{z_{h-1} = q, z_h = y\} \tag{73}$$

Next, we compute $\xi_{A,j}$ for $j \in [N+1]$. Recall that $C_{q,y} = \sum_{h=1}^H \mathbb{1}\{z_{h-1} = q, z_h = y\}$. For $j \neq \tau$, we have $\xi_{A,j} = \lambda C_{q,y} \mathbb{1}\{j = y\}$ similar to the proof of Lemma 3.1. For $j = \tau$, we have

$$\xi_{A,\tau} = \boldsymbol{e}_\tau^\top \boldsymbol{U}\boldsymbol{V} \sum_{h=1}^H (\boldsymbol{x}_H^\top W \boldsymbol{x}_h)\boldsymbol{x}_h = \boldsymbol{e}_\tau^\top \sum_{h=1}^H (\boldsymbol{x}_H^\top W \boldsymbol{x}_h)\boldsymbol{e}_{z_h} = 0.$$

We conclude that for all $j \in [N+1]$,
$$\xi_{A,j} = \lambda C_{q,y} \mathbb{1}\{j = y\} \tag{74}$$

Next, we compute $\xi_{F,j}$ for $j \in [N]$. We have $\boldsymbol{V} = \boldsymbol{I}_d$, $\boldsymbol{F} = E(\tau) \left( \sum_{q' \in \mathcal{Q}} \gamma E^\top(q') + \tilde{E}^\top(q') \right)$ and

$$\xi_F = \boldsymbol{U}\boldsymbol{F} \left( \boldsymbol{x}_H + \sum_{h=1}^H (\boldsymbol{x}_H^\top W \boldsymbol{x}_h)\boldsymbol{V}\boldsymbol{x}_h \right)$$

$$= \boldsymbol{U}E(\tau) \left( \sum_{q' \in \mathcal{Q}} \gamma E^\top(q') + \tilde{E}^\top(q') \right) \left( \boldsymbol{x}_H + \sum_{h=1}^H (\boldsymbol{x}_H^\top W \boldsymbol{x}_h)\boldsymbol{x}_h \right)$$

$$= \boldsymbol{e}_\tau \left( \sum_{q' \in \mathcal{Q}} \gamma E^\top(q') + \tilde{E}^\top(q') \right) (E(q) + \tilde{E}(z_{H-1}) + \lambda C_{q,y}(E(y) + \tilde{E}(q)))$$

$$= \boldsymbol{e}_\tau \left( \gamma E^\top(q) + \tilde{E}^\top(q) \right) (E(q) + \tilde{E}(z_{H-1}) + \lambda C_{q,y}(E(y) + \tilde{E}(q)))$$

$$= (\gamma + \lambda C_{q,y})\boldsymbol{e}_\tau,$$

where the last two equalities uses the fact that $z_{H-1}$ cannot be a trigger word, otherwise the condition IV in the data model 2.1 would be violated.

It follows that

$$\xi_{F,j} = (\gamma + \lambda C_{q,y})\mathbb{1}\{j = \tau\}. \tag{75}$$

Overall, we have

$$\xi_{A,y} = \lambda C_{q,y}, \ \xi_{A,\tau} = 0, \ \xi_{F,y} = 0, \ \xi_{F,\tau} = \gamma + \lambda C_{q,y}.$$

This implies that

- If $j = y$ then $\xi_j = \xi_{A,y} + \xi_{F,y} = \lambda C_{q,y}$.

- If $j = \tau$ then $\xi_j = \xi_{A,\tau} + \xi_{F,\tau} = \gamma + \lambda C_{q,y} = \lambda C_{q,y} + \ln \frac{\alpha}{1-\alpha}$.

- Otherwise, $\xi_j = 0$.

We conclude that $\xi_j = \mathbb{1}\{j = y \vee j = \tau\}(\lambda C_{q,y} + \mathbb{1}\{j = \tau\}\gamma)$. $\qquad\square$

**Lemma F.2.** *(Optimality of softmax attention for noisy task) By setting* $\boldsymbol{U} = [E(1) \ E(2) \ \dots \ E(N) \ E(N + 1)]^\top, \boldsymbol{V} = s\boldsymbol{I}_d, \boldsymbol{W} = \lambda \sum_{q\in\mathcal{Q}} E(q)(\tilde{E}^\top(q) - 2E^\top(\tau) - \sum_{x=1,x\neq q}^N \tilde{E}(x)^\top)$ *and* $\boldsymbol{F} = E(\tau) \sum_{q\in\mathcal{Q}}(\gamma E^\top(q) + \tilde{E}^\top(q))$, *for all* $\alpha \in (0,1)$, *using softmax attention we obtain*

$$\lim_{s\to\infty,\lambda\to\infty,\gamma\to\ln\frac{\alpha}{1-\alpha}} L(s,\gamma,\lambda) := \tag{76}$$

$$\lim_{s\to\infty} \lim_{\lambda\to\infty,\gamma\to\ln\frac{\alpha}{1-\alpha}} \mathbb{E}_{y,z_{1:H+1}}\left[-\ln\frac{\exp\big(\xi_{z_{H+1}}\big)}{\sum_{j\in[N+1]}\exp(\xi_j)}\right] = L_{\text{Bayes}}. \tag{77}$$

*Proof.* Consider a sentence with a trigger $q$ and output $y$. We have

$$\boldsymbol{x}_H^\top \boldsymbol{W} = (E(q) + \tilde{E}(z_{H-1}))^\top \lambda \sum_{q'\in\mathcal{Q}} E(q')(\tilde{E}^\top(q') - 2E^\top(\tau) - \sum_{x=1,x\neq q'}^N \tilde{E}(x)^\top) \tag{78}$$

$$= \lambda(\tilde{E}^\top(q) - 2E^\top(\tau) - \sum_{x=1,x\neq q}^N \tilde{E}(x)^\top). \tag{79}$$

It follows that

$$\boldsymbol{x}_H^\top \boldsymbol{W} \boldsymbol{x}_h = (E(q) + \tilde{E}(z_{H-1}))^\top \lambda E(q) \left(\tilde{E}^\top(q) - 2E^\top(\tau) - \sum_{x=1,x\neq q}^N \tilde{E}(x)^\top\right) \boldsymbol{x}_h \tag{80}$$

$$= \lambda \left(\tilde{E}^\top(q) - 2E^\top(\tau) - \sum_{x=1,x\neq q}^N \tilde{E}(x)^\top\right)(E(z_h) + \tilde{E}(z_{h-1})) \tag{81}$$

$$= \lambda \left(-2\mathbb{1}\{z_h = \tau\} + \mathbb{1}\{z_{h-1} = q\} - \mathbb{1}\{z_{h-1} \neq q\}\right). \tag{82}$$

Hence,

$$\boldsymbol{x}_H^\top \boldsymbol{W} \boldsymbol{x}_h = \begin{cases} \lambda & \text{if } z_{h-1} = q, z_h = y \\ -\lambda & \text{if } z_{h-1} = q, z_h = \tau \\ -\lambda & \text{otherwise.} \end{cases} \tag{83}$$

Consequently,

$$\sum_{j=1}^H \exp\big(\boldsymbol{x}_H^\top \boldsymbol{W} \boldsymbol{x}_h\big) = \sum_{z_{j-1}=q,z_j=y} \exp(\lambda) + \sum_{(z_{j-1},z_j)\neq(q,y)} \exp(-\lambda) \tag{84}$$

$$= C_{q,y} \exp(\lambda) + (H - C_{q,y}) \exp(-\lambda) \tag{85}$$

The attention score at the $h$-th token in a sentence is

$$\sigma(\boldsymbol{x}_H^\top \boldsymbol{W} \boldsymbol{x}_h) = \frac{\exp(\boldsymbol{x}_H^\top \boldsymbol{W} \boldsymbol{x}_h)}{\sum_{j=1}^H \exp(\boldsymbol{x}_H^\top \boldsymbol{W} \boldsymbol{x}_j)} \tag{86}$$

$$= \begin{cases} \frac{\exp(\lambda)}{C_{q,y}\exp(\lambda_q) + (H - C_{q,y})\exp(-\lambda)} & \text{if } z_{h-1} = q, z_h = y \\ \frac{\exp(-\lambda)}{C_{q,y}\exp(\lambda_q) + (H - C_{q,y})\exp(-\lambda)} & \text{otherwise.} \end{cases} \tag{87}$$

Next, we compute $\xi_{A,j}$. With $j = y$, we have

$$\xi_{A,y} = \boldsymbol{e}_y^\top \boldsymbol{U} \boldsymbol{V} \sum_{h=1}^H \sigma(\boldsymbol{x}_H^\top \boldsymbol{W} \boldsymbol{x}_h) \boldsymbol{x}_h = s \sum_{h=1}^H \sigma(\boldsymbol{x}_H^\top \boldsymbol{W} \boldsymbol{x}_h) \mathbb{1}\{z_h = y\} \tag{88}$$

$$= s \left( \sum_{z_{h-1}=q} \sigma(\boldsymbol{x}_H^\top \boldsymbol{W} \boldsymbol{x}_h) \mathbb{1}\{z_h = y\} + \sum_{z_{h-1}\neq q} \sigma(\boldsymbol{x}_H^\top \boldsymbol{W} \boldsymbol{x}_h) \mathbb{1}\{z_h = y\} \right) \tag{89}$$

$$= s \frac{C_{q,y}\exp(\lambda) + (C_y - C_{q,y})\exp(-\lambda)}{C_{q,y}\exp(\lambda) + (H - C_{q,y})\exp(-\lambda)}. \tag{90}$$

With $j \neq y$, we have

$$\xi_{A,j} = \boldsymbol{e}_j^\top \boldsymbol{U} \boldsymbol{V} \sum_{h=1}^H \sigma(\boldsymbol{x}_H^\top \boldsymbol{W} \boldsymbol{x}_h) \boldsymbol{x}_h = s \sum_{h=1}^H \sigma(\boldsymbol{x}_H^\top \boldsymbol{W} \boldsymbol{x}_h) \mathbb{1}\{z_h = j\} \tag{91}$$

$$= s \sum_{z_{h-1}\neq q} \frac{\exp(-\lambda)}{C_{q,y}\exp(\lambda_q) + (H - C_{q,y})\exp(-\lambda)} \mathbb{1}\{z_h = j\} \tag{92}$$

$$= s \frac{C_j \exp(-\lambda)}{C_{q,y}\exp(\lambda) + (H - C_{q,y})\exp(-\lambda_q)}. \tag{93}$$

Next, the logits of the feed-forward layer is

$$\xi_F = \boldsymbol{U} \boldsymbol{F} \left( \boldsymbol{x}_H + \sum_{h=1}^H \sigma(\boldsymbol{x}_H^\top \boldsymbol{W} \boldsymbol{x}_h) \boldsymbol{V} \boldsymbol{x}_h \right)$$

$$= \boldsymbol{U} E(\tau) \left( \sum_{q'\in\mathcal{Q}} \gamma E^\top(q') + \tilde{E}^\top(q') \right) \left( E(q) + \tilde{E}(z_{H-1}) + s \sum_{h=1}^H \sigma(\boldsymbol{x}_H^\top \boldsymbol{W} \boldsymbol{x}_h) \boldsymbol{x}_h \right)$$

$$= \boldsymbol{e}_\tau \left( \gamma + s \sum_{q'\in\mathcal{Q}} \gamma \sum_{h=1}^H \sigma(\boldsymbol{x}_H^\top \boldsymbol{W} \boldsymbol{x}_h) E^\top(q') \boldsymbol{x}_h + \sum_{h=1}^H \sigma(\boldsymbol{x}_H^\top \boldsymbol{W} \boldsymbol{x}_h) \tilde{E}^\top(q') \boldsymbol{x}_h \right)$$

$$= \boldsymbol{e}_\tau \left( \gamma + s \sum_{q'\in\mathcal{Q}} \gamma \sum_{h=1}^H \sigma(\boldsymbol{x}_H^\top \boldsymbol{W} \boldsymbol{x}_h) \mathbb{1}\{z_h = q'\} + \sum_{h=1}^H \sigma(\boldsymbol{x}_H^\top \boldsymbol{W} \boldsymbol{x}_h) \mathbb{1}\{z_{h-1} = q'\} \right)$$

$$= \boldsymbol{e}_\tau \left( \gamma + \left( s \sum_{q'\in\mathcal{Q}} \frac{\gamma C_{q'} \exp(-\lambda)}{C_{q,y}\exp(\lambda) + (H - C_{q,y})\exp(-\lambda)} \right) \right)$$

$$+ \boldsymbol{e}_\tau \left( s \frac{C_{q,y}\exp(\lambda) + C_\tau \exp(-\lambda)}{C_{q,y}\exp(\lambda) + (H - C_{q,y})\exp(-\lambda)} + \sum_{q'\in\mathcal{Q}, q'\neq q} \frac{C_{q'} \exp(-\lambda)}{C_{q,y}\exp(\lambda) + (H - C_{q,y})\exp(-\lambda)} \right),$$

where we used $E^\top(q')\boldsymbol{x}_h = \mathbb{1}\{z_h = q'\}$ and $\tilde{E}^\top(q')\boldsymbol{x}_h = \mathbb{1}\{z_{h-1} = q'\}$. As a result, the combined logits of attention and feed-forward layers is

- If $j = y$, then

$$\xi_y = \xi_{A,y} + \xi_{F,y} = \xi_{A,y} \tag{94}$$

$$= s \frac{C_{q,y}\exp(\lambda) + (C_y - C_{q,y})\exp(-\lambda)}{C_{q,y}\exp(\lambda) + (H - C_{q,y})\exp(-\lambda)}. \tag{95}$$

It follows that $\lim_{\lambda\to\infty} \xi_y = s$.

- If $j = \tau$, then

$$\xi_\tau = \xi_{A,\tau} + \xi_{F,\tau} \tag{96}$$

$$= s\frac{2C_\tau \exp(-\lambda) + \gamma C_q \exp(-\lambda)}{C_{q,y}\exp(\lambda_q) + (H - C_{q,y})\exp(-\lambda_q)} + \gamma + s\frac{C_{q,y}\exp(\lambda)}{C_{q,y}\exp(\lambda) + (H - C_{q,y})\exp(-\lambda)} \tag{97}$$

$$+ \sum_{q'\in\mathcal{Q}, q'\neq q} \frac{C_{q'}\exp(-\lambda)}{C_{q,y}\exp(\lambda) + (H - C_{q,y})\exp(-\lambda)}. \tag{98}$$

It follows that $\lim_{\lambda\to\infty}\xi_\tau = s + \gamma$.

- Otherwise, $\xi_j = \xi_{A,j} + \xi_{F,j} = \xi_{A,j} = s\frac{C_j\exp(-\lambda)}{C_{q,y}\exp(\lambda)+(H-C_{q,y})\exp(-\lambda_q)}$.

It follows that $\lim_{\lambda\to\infty}\xi_j = 0$.

Then, Lemma F.2 follows directly from

$$\lim_{s\to\infty}\lim_{\lambda\to\infty}\frac{\exp(\xi_y)}{\exp(\xi_y)+\exp(\xi_\tau)+\sum_{x=1,x\neq y}^N \exp(\xi_x)} = \lim_{s\to\infty}\frac{\exp(s)}{\exp(s)+\exp(s+\gamma)+N-1}$$

$$= \lim_{s\to\infty}\frac{1}{1+\exp(\gamma)+(N-1)\exp(-s)}$$

$$= \frac{1}{1+\exp(\gamma)}$$

$$= 1-\alpha \quad \text{as } \gamma \to \ln\left(\frac{\alpha}{1-\alpha}\right),$$

and

$$\lim_{s\to\infty}\lim_{\lambda\to\infty}\frac{\exp(\xi_\tau)}{\exp(\xi_y)+\exp(\xi_\tau)+\sum_{x=1,x\neq y}^N \exp(\xi_x)} = \lim_{s\to\infty}\frac{\exp(s+\gamma)}{\exp(s)+\exp(s+\gamma)+N-1}$$

$$= \lim_{s\to\infty}\frac{\exp(\gamma)}{1+\exp(\gamma)+(N-1)\exp(-s)}$$

$$= \frac{\exp(\gamma)}{1+\exp(\gamma)}$$

$$= \alpha \quad \text{as } \gamma \to \ln\left(\frac{\alpha}{1-\alpha}\right).$$

$\square$

### F.2 ANALYSIS OF NORMALIZED GRADIENT DESCENT ON POPULATION AND EMPIRICAL LOSSES

To avoid notational overload, we write $C_{q,y}^{(m)}$ for $C_{q,y^{(m)}}^{(m)}$. We first consider the case where $\alpha$ is known, and then extend the analysis to the case of unknown $\alpha$.

#### F.2.1 KNOWN $\alpha$: RUNNING NORMALIZED GRADIENT DESCENT ON THE POPULATION LOSS $L(\lambda)$

We will drop the subscript in $C_{q,y}$ and just write $C$ when it is referring to a generic $q, y$ under the expectation sign. Set $\gamma = \ln\frac{\alpha}{1-\alpha}$ and let $\beta = C\lambda + \gamma$. The population loss in the noisy learning

---

**Algorithm 1** Finite-Sample Training Algorithm with unknown $\alpha$

---

**Input:** $M$ i.i.d sentences $(z_{1:H+1}^{(m)})_{m=1,2,\ldots,M}$, learning rate $\eta > 0$
Compute $M_\tau = \sum_{m=1}^{M} \mathbb{1}\{z_{H+1}^{(m)} = \tau\}$
Compute $\hat{\alpha} = \frac{M_\tau}{M}$ and $\hat{\gamma} = \ln \frac{\hat{\alpha}}{1-\hat{\alpha}}$
Initialize $\lambda_0 = 0$
**for** each round $t = 1, \ldots,$ **do**
 Compute $C_{q,y}^{(m)} = \sum_{h=1}^{H-1} \mathbb{1}\{z_{h-1}^{(m)} = q, z_h^{(m)} = y\}$
 Compute $L_{\text{emp}} = \frac{1}{M}\left(\sum_{m=1}^{M} -C_{q,y}^{(m)}\lambda + \ln\left(\frac{\exp(C_{q,y}^{(m)}\lambda)}{1-\hat{\alpha}} + N - 1\right)\right) + \frac{M_\tau}{M}\ln\frac{\hat{\alpha}}{1-\hat{\alpha}}$
 Update $\lambda_t = \lambda_{t-1} - \eta \frac{dL_{\text{emp}}}{d\lambda_t}$

---

setting is defined as

$$
L_{\text{pop}}(\lambda, \gamma) = \mathbb{E}\left[(1-\alpha)\left(-\ln\frac{e^{C\lambda}}{e^{C\lambda} + e^\beta + N - 1}\right) + \alpha\left(-\ln\frac{e^\beta}{e^{C\lambda} + e^\beta + N - 1}\right)\right]
$$

$$
= \mathbb{E}\left[(1-\alpha)(-C\lambda) - \alpha\beta + \ln(e^{C\lambda} + e^\beta + N - 1)\right]
$$

$$
= \mathbb{E}\left[(\alpha-1)C\lambda - \alpha(C\lambda + \ln\frac{\alpha}{1-\alpha}) + \ln\left(e^{C\lambda} + e^{C\lambda}\frac{\alpha}{1-\alpha} + N - 1\right)\right]
$$

$$
= \mathbb{E}\left[-C\lambda + \ln\left(\frac{e^{C\lambda}}{1-\alpha} + N - 1\right) - \alpha\ln\frac{\alpha}{1-\alpha}\right].
$$

By Lemma G.1, the derivative of $f(\lambda) = -C\lambda + \ln\left(\frac{e^{C\lambda}}{1-\alpha} + N - 1\right)$ is negative. Hence, $\frac{dL_{\text{pop}}}{d\lambda} < 0$. It follows that running normalized gradient descent on $L_{\text{pop}}(\lambda)$ from $\lambda_0 = 0$ gives

$$
\lambda_t = \lambda_{t-1} - \eta\frac{\frac{dL_{\text{pop}}}{d\lambda}}{\left|\frac{dL_{\text{pop}}}{d\lambda}\right|} = \lambda_{t-1} + \eta = \eta t. \tag{99}
$$

It follows that

$$
L_{\text{pop}}(\lambda_t, \gamma) = \mathbb{E}\left[-C_{q,y}\lambda_t + \ln\left(\frac{e^{C_{q,y}\lambda_t}}{1-\alpha} + N - 1\right) - \alpha\ln\frac{\alpha}{1-\alpha}\right]
$$

$$
= \mathbb{E}\left[\ln\left(e^{-C_{q,y}\lambda_t}\left(\frac{e^{C_{q,y}\lambda_t}}{1-\alpha} + N - 1\right)\right) - \alpha\ln\frac{\alpha}{1-\alpha}\right]
$$

$$
= \mathbb{E}\left[\ln\left(\frac{1}{1-\alpha} + e^{-C_{q,y}\eta t}(N-1)\right) - \alpha\ln\frac{\alpha}{1-\alpha}\right]
$$

$$
\leq \mathbb{E}\left[\ln\left(\frac{1}{1-\alpha} + e^{-\eta t}(N-1)\right) - \alpha\ln\frac{\alpha}{1-\alpha}\right]
$$

$$
\leq -\alpha\ln\alpha - (1-\alpha)\ln(1-\alpha) + (N-1)e^{-\eta t},
$$

where the last two inequalities are from $C_{q,y} \geq 1$ and

$$
\ln\left(\frac{1}{1-\alpha} + e^{-\eta t}(N-1)\right) = \ln\left(\frac{1}{1-\alpha}\right) + \ln\left(1 + (1-\alpha)(N-1)e^{-\eta t}\right)
$$

$$
\leq \ln\left(\frac{1}{1-\alpha}\right) + (N-1)e^{-\eta t}
$$

due to $\ln(1+x) \leq x$ for all $x \geq -1$.

F.2.2  UNKNOWN $\alpha$: PROOF OF THEOREM 5.4

*Proof.* Let $\beta^{(m)} = C_{q,y}^{(m)}\lambda + \ln\frac{\hat{\alpha}}{1-\hat{\alpha}}$. The empirical loss is

$$L_{\text{emp}}(\lambda) = \frac{1}{M}\sum_{m=1}^{M} -\ln\frac{\exp\left(\xi_{z_{H+1}}^{(m)}\right)}{\sum_{j\in[N+1]}\exp\left(\xi_j^{(m)}\right)}$$

$$= \frac{1}{M}\sum_{m=1}^{M} -\xi_{z_{H+1}}^{(m)} + \ln\left(\sum_{j\in[N+1]}\exp\left(\xi_j^{(m)}\right)\right)$$

$$= \frac{1}{M}\left(\sum_{m=1,z_{H+1}^{(m)}=\tau}^{M} -\xi_\tau^{(m)} + \sum_{m=1,z_{H+1}^{(m)}\neq\tau}^{M} -\xi_y^{(m)} + \sum_{m=1}^{M}\ln\left(\sum_{j\in[N+1]}\exp\left(\xi_j^{(m)}\right)\right)\right).$$

Using $\xi_\tau^{(m)} = \beta^{(m)} = C_{q,y}^{(m)}\lambda + \ln\frac{\hat{\alpha}}{1-\hat{\alpha}}, \xi_y^{(m)} = C_{q,y}^{(m)}\lambda$ and $\xi_j^{(m)} = 0$ for $j\notin\{q,y\}$, we obtain

$$L_{\text{emp}}(\lambda) = \frac{1}{M}\left(\sum_{m=1,z_{H+1}^{(m)}=\tau}^{M}\beta^{(m)} + \sum_{m=1,z_{H+1}^{(m)}\neq\tau}^{M} -C_{q,y}^{(m)}\lambda + \sum_{m=1}^{M}\ln\left(\exp\left(\beta^{(m)}\right) + \exp\left(C_{q,y}^{(m)}\lambda\right) + N - 1\right)\right)$$

$$= \frac{1}{M}\left(\sum_{m=1}^{M} -C_{q,y}^{(m)}\lambda + \ln\left(\exp\left(\beta^{(m)}\right) + \exp\left(C_{q,y}^{(m)}\lambda\right) + N - 1\right)\right) + \frac{M_\tau}{M}\ln\frac{\hat{\alpha}}{1-\hat{\alpha}}$$

$$= \frac{1}{M}\left(\sum_{m=1}^{M} -C_{q,y}^{(m)}\lambda + \ln\left(\frac{\exp\left(C_{q,y}^{(m)}\lambda\right)}{1-\hat{\alpha}} + N - 1\right)\right) + \frac{M_\tau}{M}\ln\frac{\hat{\alpha}}{1-\hat{\alpha}}.$$

By Lemma G.1, we have $\frac{dL_{\text{emp}}}{d\lambda} < 0$. As a result, running normalized gradient descent on $L_{\text{emp}}$ gives $\lambda_t = \eta t$. The population loss is

$$L_{\text{pop}}(\lambda_t, \hat{\gamma}) = \mathbb{E}\left[(1-\alpha)\left(-\ln\frac{e^{C_{q,y}\lambda_t}}{e^{C_{q,y}\lambda_t} + e^{C_{q,y}\lambda_t+\hat{\gamma}_t} + N - 1}\right) + \alpha\left(-\ln\frac{e^{C_{q,y}\lambda_t+\hat{\gamma}_t}}{e^{C_{q,y}\lambda_t} + e^{C\lambda+\hat{\gamma}_t} + N - 1}\right)\right]$$

$$= \mathbb{E}\left[-C_{q,y}\lambda_t + \ln\left(e^{C_{q,y}\lambda_t} + e^{C_{q,y}\lambda_t+\hat{\gamma}_t} + N - 1\right) - \alpha\hat{\gamma}\right]$$

$$= \mathbb{E}\left[-C_{q,y}\lambda_t + \ln\left(\frac{e^{C_{q,y}\lambda_t}}{1-\hat{\alpha}} + N - 1\right)\right] - \alpha\hat{\gamma}$$

$$= \mathbb{E}\left[\ln\left(\frac{1}{1-\hat{\alpha}} + e^{-C_{q,y}\eta t}(N - 1)\right)\right] - \alpha\ln\frac{\hat{\alpha}}{1-\hat{\alpha}}$$

$$\leq \ln\left(\frac{1}{1-\hat{\alpha}} + e^{-\eta t}(N - 1)\right) - \alpha\ln\frac{\hat{\alpha}}{1-\hat{\alpha}}$$

$$\leq -\alpha\ln\hat{\alpha} - (1-\alpha)\ln(1-\hat{\alpha}) + (N-1)e^{-\eta t}$$

$$= -\alpha\ln\alpha - (1-\alpha)\ln(1-\alpha) + KL(\alpha\parallel\hat{\alpha}) + (N-1)e^{-\eta t}$$

$$= L_{\text{Bayes}} + KL(\alpha\parallel\hat{\alpha}) + (N-1)e^{-\eta t},$$

where $KL(\alpha\parallel\hat{\alpha})$ is the Kullback-Leibler divergence between two Bernoulli distributions $\text{Ber}(\alpha)$ and $\text{Ber}(\hat{\alpha})$. By Lemma G.2, we have with probability at least $1-\delta$,

$$L_{\text{pop}}(\lambda_t, \hat{\gamma}) \leq L_{\text{Bayes}} + \frac{1}{\min(\alpha, 1-\alpha) - \sqrt{\frac{\ln(2/\delta)}{2M}}}\frac{\ln(2/\delta)}{2M} + (N-1)e^{-\eta t}.$$

$\square$

### F.3 PROOF OF THEOREM 5.5

*Proof.* By Equation (66), we have

$$\xi_j = \mathbb{1}\{j = y_{\text{test}} \vee j = \tau\}(\lambda_t C_{q,y_{\text{test}}} + \mathbb{1}\{j = \tau\}\hat{\gamma}). \tag{100}$$

Using $\hat{\gamma} = \ln \frac{\hat{\alpha}}{1-\hat{\alpha}}$, we obtain

$$\Pr[z_{H+1} = y_{\text{test}} \mid \lambda_t, \hat{\gamma}] := \frac{\exp(\xi_{y_{\text{test}}})}{\sum_{j \in [N+1]} \exp(\xi_j)} \tag{101}$$

$$= \frac{\exp(\lambda_t C_{q,y_{\text{test}}})}{\exp(\lambda_t C_{q,y_{\text{test}}}) + \exp(\lambda_t C_{q,y_{\text{test}}} + \hat{\gamma}) + N - 1} \tag{102}$$

$$= \frac{\exp(C_{q,y} \lambda_t)}{\frac{\exp(C_{q,y_{\text{test}}} \lambda_t)}{1-\hat{\alpha}} + N - 1} \tag{103}$$

$$= (1 - \hat{\alpha}) \frac{1}{1 + (N-1)(1-\hat{\alpha}) e^{-C_{q,y_{\text{test}}} \lambda_t}} \tag{104}$$

$$= (1 - \hat{\alpha}) \frac{1}{1 + (N-1)(1-\hat{\alpha}) e^{-C_{q,y_{\text{test}}} \eta t}} \tag{105}$$

For large $t$, the quantity $e^{-C_{q,y_{\text{test}}} \eta t}$ is close to 0. By Taylor's theorem, we have $\frac{1}{1+x} = 1 - x + O(x^2)$ for small $x$. Therefore, with probability at least $1 - \delta$,

$$\Pr[z_{H+1} = y_{\text{test}} \mid \lambda_t, \hat{\gamma}] = (1 - \hat{\alpha}) \left( 1 - (N-1)(1-\hat{\alpha}) e^{-C_{q,y_{\text{test}}} \eta t} + O(N^2 e^{-2C_{q,y_{\text{test}}} \eta t}) \right) \tag{106}$$

$$= 1 - \hat{\alpha} + O(N^2 e^{-2\eta t}) \tag{107}$$

$$= 1 - \alpha + O\left( \sqrt{\frac{\ln(1/\delta)}{M}} + N^2 e^{-2\eta t} \right), \tag{108}$$

where the last equality is from $\hat{\alpha} = \alpha - O(\sqrt{\frac{\ln(1/\delta)}{M}})$ with probability at least $1 - \delta$.

The proof for $\Pr[z_{H+1} = \tau \mid \lambda_t, \hat{\gamma}]$ follows similarly. $\qquad\square$

### F.4 PROOF OF THEOREM 5.6

*Proof.* Recall that $\hat{\gamma} = \ln \frac{1-\hat{\alpha}}{\hat{\alpha}}$. By Hoeffding's inequality, we have $|\alpha - \hat{\alpha}| \leq \sqrt{\frac{\ln(2/\delta)}{2M}}$ with probability at least $1 - \delta$. Hence, $t \geq \max(1, \frac{1}{\eta} | \ln \left( 1 - \alpha + \sqrt{\frac{\ln(2/\delta)}{2M}} \right) - \ln \left( \alpha - \sqrt{\frac{\ln(2/\delta)}{2M}} \right) |)$ implies that $t \geq \max(1, -\hat{\gamma}/\eta)$.

By Equation (74), we have

$$\xi_{A,y} = \lambda_t C_{q,y} = \eta t C_{q,y} > 0 = \max_{j \neq y} \xi_{A,j}. \tag{109}$$

By Equation (75), we have

$$\xi_{F,\tau} = \lambda_t C_{q,y} + \hat{\gamma} \geq \eta t + \hat{\gamma} > 0 = \max_{j \neq \tau} \xi_{F,j} \tag{110}$$

since $C_{q,y} \geq 1$ and $\lambda_t = \eta t > \max(0, -\hat{\gamma})$ for $t \geq \max(1, -\frac{\hat{\gamma}}{\eta})$. We conclude that the condition (4) is satisfied with probability at least $1 - \delta$. $\qquad\square$

## G TECHNICAL LEMMAS

**Lemma G.1.** *For any $N > 1, \alpha \in [0,1), C > 0$, the derivative of the function*

$$f(x) = -Cx + \ln \left( \frac{\exp(Cx)}{1-\alpha} + N - 1 \right)$$

*is negative for all $x \in \mathbb{R}$.*

*Proof.* We have $\frac{df}{dx} = -C + \frac{\frac{Ce^{Cx}}{1-\alpha}}{\frac{\exp(Cx)}{1-\alpha} + N - 1} = \frac{C(1-N)}{\frac{\exp(Cx)}{1-\alpha} + N - 1} < 0$. $\qquad\square$

**Lemma G.2.** *Let $\alpha \in (0,1)$. Let $M$ i.i.d samples $(X_i)_{i \in [M]}$ be drawn from $X_i \sim \mathrm{Ber}(\alpha)$. Let $\hat{\alpha} = \frac{1}{M} \sum_{i=1}^{M} X_i$. With probability at least $1 - \delta$, we have*

$$KL(\alpha \parallel \hat{\alpha}) \leq \frac{1}{\min(\alpha, 1-\alpha) - \sqrt{\frac{\ln(2/\delta)}{2M}}} \frac{\ln(2/\delta)}{2M}.$$

*Proof.* By the reverse Pinsker's inequality (Sason, 2015, Theorem 3), we have

$$KL(\alpha \parallel \hat{\alpha}) \leq \frac{2(\alpha - \hat{\alpha})^2}{\min(\hat{\alpha}, 1-\hat{\alpha})}.$$

By Hoeffding's inequality, the event $|\alpha - \hat{\alpha}| \leq \sqrt{\frac{\ln(2/\delta)}{2M}}$ holds with probability at least $1 - \delta$. Under this event, we have $(\alpha - \hat{\alpha})^2 \leq \frac{\ln(2/\delta)}{2M}$. Also, $\hat{\alpha} \geq \alpha - \sqrt{\frac{\ln(2/\delta)}{2M}}$ and $1 - \hat{\alpha} \geq 1 - \alpha - \sqrt{\frac{\ln(2/\delta)}{2M}}$. Hence, $\min(\hat{\alpha}, 1-\hat{\alpha}) \geq \min(\alpha, 1-\alpha) - \sqrt{\frac{\ln(2/\delta)}{2M}}$. The statement follows immediately.

$\square$

# H  FURTHER DETAILS ON EXPERIMENTS

## H.1  ADDITIONAL DETAILS ON THE EXPERIMENTAL SETUP

**Hyperparameters** The majority of our experiments are repeated five times with five random seeds from $0$ to $5$. However, possibly due to the large number of iterations and large size of the finite dataset, no significant differences are observed between different random seeds.

We also experimented with several different values of learning rates ranging from $0.1$ to $0.8$. Consistent with the theoretical findings, we find that the more reparamterized a model is, the less sensitive it is to changes in the learning rate. All of our results are reported for learning rates set at either $0.1, 0.2$ or $0.8$.

In finite-sample experiments, we train the models on a dataset of size $M = 2048$ samples and then compute the models' population losses and unseen output test losses. To calculate the population loss, we use a freshly sampled dataset of size $10M = 20480$ samples. To calculate the unseen output test losses, we use a freshly sampled dataset of size $512$ samples, where $y \in [1, 4]$ is replaced by a randomly chosen $y_{\text{test}} \in [5, 59]$.

**Computing Resources** The experiments are implemented in PyTorch. All experiments are run on a single-CPU computer. The processor is `11th Gen Intel(R) Core i7-11700K` with 32 GB RAM. Training all models simultaneously takes about 30 minutes from start to finish.

## H.2  ATTENTION LAYER LEARNS TO PREDICT OUTPUT TOKENS WHILE FEED-FORWARD LAYER LEARNS TO PREDICT NOISE TOKEN

To measure the extent to which we can separate the learning functionality of the attention layer and the feed-forward layer, we train three models `Origin-Linear`, `Reparam-Linear` and `Reparam-Linear-W`, and record the logits of each layer on the output tokens, the noise tokens and the maximum values in the logits of the two layers on all three noisy tasks with $\alpha = 0.2, 0.5$ and $0.8$. We use $\eta = 0.1$ in all experiments. The results are reported in Figures 3 to 5.

We say that the attention layer and the feed-forward layer learns to predict the output and noise tokens, respectively, if $\xi_{A,y} = \max_j \xi_{A,j}$ and $\xi_{F,\tau} = \max_j \xi_{F,\tau}$. It can be observed that all three models exhibit some layer-specific learning mechanism, including the original model where all three matrices $V, W$ and $F$ are trained from scratch without reparameterization. However, depending on the noise level, at least one of the two layers in the original model do not fully specialize in either the output nor the noise tokens. For $\alpha \leq 0.5$, Figures 3a and 3d show that the attention layer in `Origin-Linear` learns to predict output tokens perfectly, however the feed-forward layer does not always predict $\tau$.

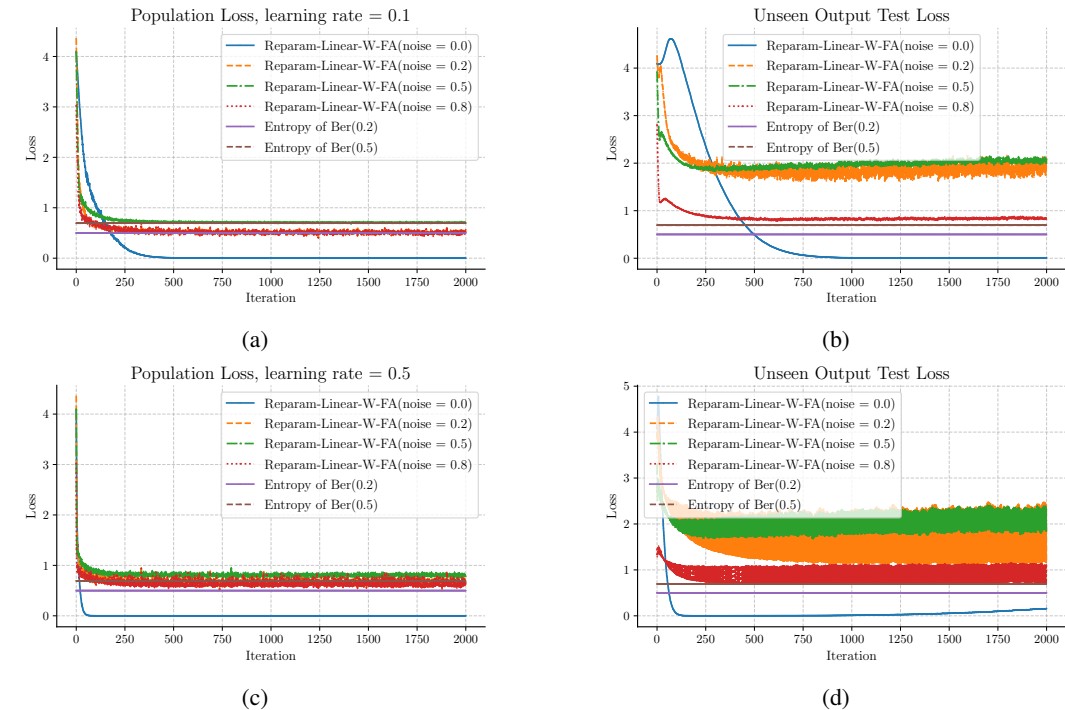

Figure 2: Population and Unseen Output Test Losses of `Reparam-Linear-W` with $\eta = 0.1$ (first row), $\eta = 0.5$ (second row). Population losses converge with limited generalization to unseen output words.

At $\alpha = 0.8$, the feed-forward layer succeeds in learning to predict $\tau$ but the attention layer fails to focus entirely on the output tokens.

In contrast, the fully-reparameterized model `Reparam-Linear` exhibits perfect separation in the functionality of the two layers, which verified our Theorem 5.6. The same phenomenon is also observed in `Reparam-Linear-W`, which suggests that reparameterizing $\boldsymbol{F}$ is sufficient to force the two layers to be biased towards two different types of tokens.

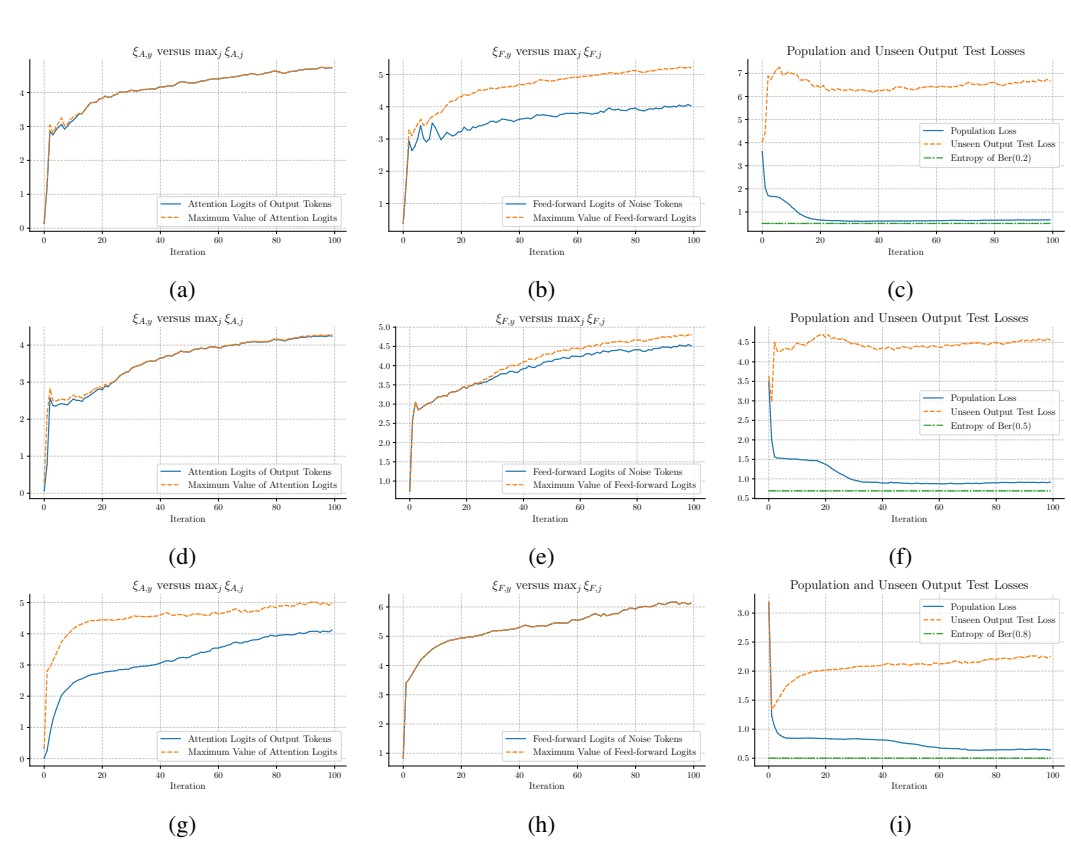

Figure 3: `Origin-Linear` with $\alpha = 0.2$ (first row), $\alpha = 0.5$ (second row) and $\alpha = 0.8$ (third row). The learning rate is $\eta = 0.1$.

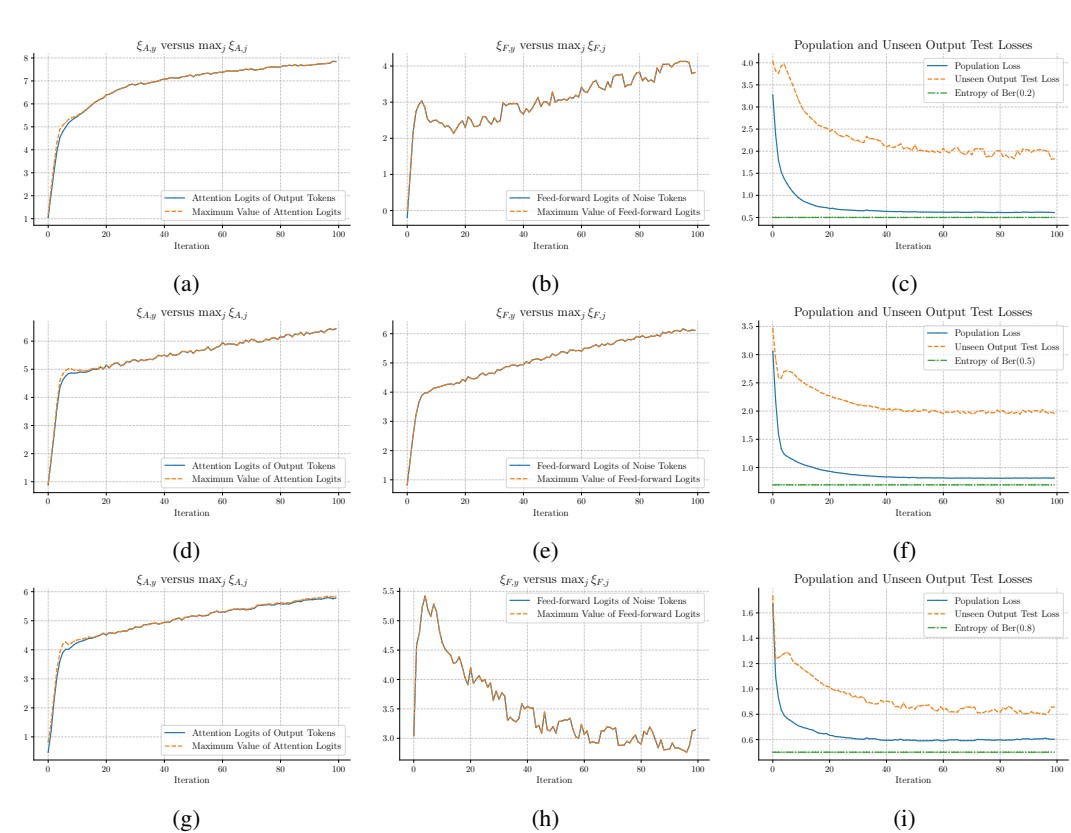

Figure 4: `Reparam-Linear-W` with $\alpha = 0.2$ (first row), $\alpha = 0.5$ (second row) and $\alpha = 0.8$ (third row). The learning rate is $\eta = 0.1$.

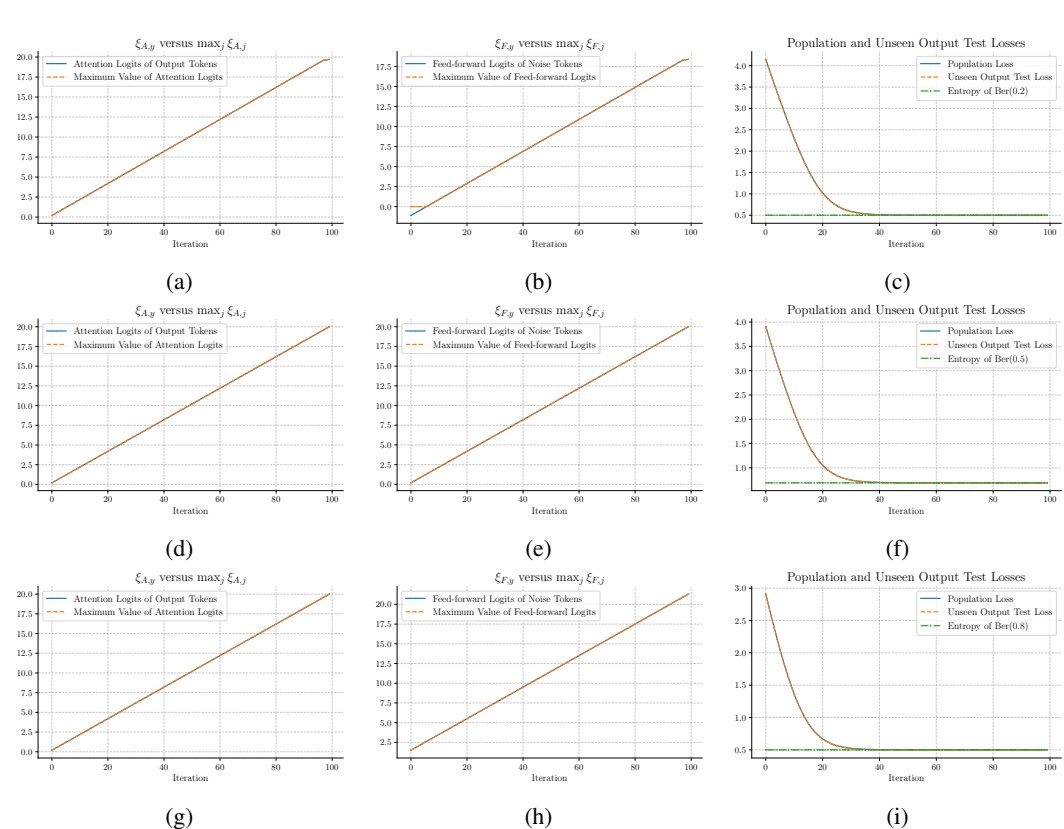

Figure 5: `Reparam-Linear` with $\alpha = 0.2$ (first row), $\alpha = 0.5$ (second row) and $\alpha = 0.8$ (third row). The learning rate is $\eta = 0.1$.

