# OpenReview forum: "How Transformers Learn In-Context Recall Tasks? Optimality, Training Dynamics and Generalization"
_ICLR.cc/2026/Conference — Submitted to ICLR 2026_

### Official Review · Reviewer_e9iq · 2025-10-25

**Soundness:** 3
**Presentation:** 1
**Contribution:** 1
**Rating:** 2
**Confidence:** 3

**Summary:**

The authors study a simple model of associative recall. They show how a simplified Transformer model may solve a recall task, but demonstrate empirically that a model trained from scratch does not generalize on this task unless parameterized in a specific way.

**Strengths:**

The authors study a timely and fascinating topic. Particularly as LLMs make their way into more and more aspects of our lives, understanding their basic capabilities is important.

**Weaknesses:**

It remains unclear to me how novel the authors' contribution is. I'm not intimately familiar with this literature, but it seems to me many of the basic results the authors present have already been discovered and analyzed extensively before?

Specifically, the central claim seems to be that transformers solve an in-context recall task optimally. This seems to have been well studied and validated since Bietti et al (https://arxiv.org/abs/2306.00802). It seems the authors seem to claim novelty by asserting that their analysis studies multiple occurrences of multiple query tokens, but I'm unsure if Bietti et al's analysis is invalidated in this setting? See also Chan et al (https://arxiv.org/abs/2410.23042), which explicitly study the role of having multiple query tokens on associative recall. Also Reddy (https://arxiv.org/abs/2312.03002, https://arxiv.org/abs/2412.00104) performs a detailed theoretical analysis of this phenomenon. Many of these studies also consider softmax attention with noise. Also, the convergence result asserting a linear rate seems to be shown already in Huang et al (https://arxiv.org/abs/2409.17335), no? Is their result invalid in your setting?

Separately, the empirics demonstrating that a Transformer trained from scratch *fails* to generalize well on your task seems to be a severe weakness. LLMs are able to solve associative recall tasks presumably without your parametrizations. Wouldn't this suggest that the phenomena you characterize with your parametrizations are not reflective of actual models?

A small aside, there seem to be frequent accidentally-omitted-words and grammatical typos that hinder your manuscript's clarity. You draft may benefit from a careful read-through to catch these typos.

**Questions:**

See Weaknesses above.

---

> ### Author Response · Authors · 2025-11-21
>
> > This seems to have been well studied and validated since Bietti et al.
>
> The short answer is no,  our central contribution is the theoretical investigations into *how* transformers solve in-context recall tasks. Our theoretical investigations include optimality, convergence rates, on-convergence behaviors, and in- and out-of-distribution generalization bounds of transformers trained on in-context recall tasks. None of these investigations were studied in Bietti et al.
>
> In particular, the central theoretical results in Bietti et al. are their Lemma 1, Lemma 2 and Theorem 3. Their Lemmas 1 and 2 consider a very simple linear model with convex objective that does not involve any attention mechanism. Their Theorem 3 holds for sequential one-step gradient update on the population loss for the output, key and value matrices in that order. The nature of this claim is completely different from our work, and also their sequential GD training on the matrices in that order seem unnatural.
>
> >... but I'm unsure if Bietti et al's analysis is invalidated?
>
>  We would like to reiterate the major differences between our task and Bietti et al. as follows (note that we have discussed these differences in our paper -- Section 2 and Remark 5.3)
> - Our task considers *noisy* output problems, where the output may be a noise token instead of a proper output. Bietti et al. consider *noiseless* problems only.
> - Our task tests the model on *unseen, out-of-distribution* samples. Bietti et al. does not study out-of-distribution samples.
> - Our analysis is both distribution-agnostic (for training on population samples) and finite-sample robust (for training on a finite dataset). Bietti et al.'s analysis uses specific, explicitly defined distributions (see their page 16, the first paragraph in appendix B.3), and does not consider finite-sample analysis.
>
> We agree that a more thorough discussion would improve the clarity of our paper. We will soon upload a new version with this discussion.
>
> > Chan et al, which explicitly study the role of having multiple query tokens on associative recall.
>
> From our examination, Chan et al does not study associative recall. In particular, Chan et al study in-context classification problems  with high-dimensional input $x \in R^d$, which are completely different from our associative recall task.
> Could the reviewer please elaborate what exactly "the role of having multiple query tokens on associative recall" in Chan et al that you were referring to?
>
> > ... Reddy performs a detailed theoretical analysis ....
>
> We have examined both of these papers from Reddy's group closely and could not find any theoretical results presented as a formal mathematical lemma or theorem. The second paper provided a few formula for the approximation of the gradients of a loss function, however these formula alone do not constitute a formal statement on the convergence or generalization of a model. Could the reviewer please elaborate what exactly "a detailed theoretical analysis of this phenomenon" in these papers that you were referring to?
>
> > the convergence result seems to be shown already in Huang et al. (arxiv:2409.17335)?
>
> We highlight the fundamental differences between our convergence rate results and of Huang et al.:
> - One-stage versus Two-stage training procedure: in our work, we train the (parameterized) key-query and value matrices simultaneously, which can also be seen as a one-stage training.
> In contrast, Huang et al. (their Algorithm 1) follow a two-stage training first train the value matrix for $T$ rounds, only then they train the key-query matrix.
> - The absence of a hard-margin sub-problem: a key mechanism leading to the linear convergence rate in Huang et al. is the presence of a hard-margin sub-problem (see their Equation 2). This sub-problem arises out of their assumption that there exists a collocation (i.e., a one-to-one mapping) in their training sample. This assumption, and hence the hard-margin sub-problem, does not exist in our work.
>
> While we did cite and briefly discussed both of these differences in our paper, we will add a new section detailing the differences in our linear convergence rates and the one in Huang et al. in the next version (to be uploaded soon).
>
>
> > Separately, the empirics demonstrating that ...
>
> Our experiments aim to compare non-parameterized versus parameterized *one-layer* transformers.
> Our results indicate that with the same model complexity (i.e. depth and width), re-parameterization can give significant benefits in terms of sample complexity and out-of-distribution generalization guarantees.
> We do not aim to compare our parameterized one-layer models with non-parameterized large-scale models because large-scale models have significantly larger complexity, and hence, they can pre-encode a much larger space of solution biases, some of which may already be beneficial for in-context recall task.
>
> > typos
>
> Thanks, we will soon upload new version that fix all of these typos.

---

### Official Review · Reviewer_xnEr · 2025-11-01

**Soundness:** 3
**Presentation:** 3
**Contribution:** 3
**Rating:** 6
**Confidence:** 4

**Summary:**

This paper provides a theoretical and empirical analysis of how one-layer transformers learn in-context recall tasks—synthetic settings requiring recognition of positional token associations. The authors show (i) that such transformers are Bayes-optimal for both noiseless and noisy recall tasks, (ii) that gradient descent achieves linear convergence to the Bayes risk, and (iii) that the learned representations can generalize out-of-distribution (OOD) to unseen tokens.

**Strengths:**

1. This paper extends prior studies that only analyzed the first training step or infinite-sample limits by providing finite-sample analyses, explicit reparameterizations, and empirical validations demonstrating when proper parameterization is crucial for OOD generalization

2. This paper theoretically characterizes the different behaviors of the feed-forward layer and the attention layer in in-context recall tasks, which is insightful.

**Weaknesses:**

1. Limited architecture depth: Results are confined to one-layer, single-head transformers, far from the multi-layer, residual, or multi-head dynamics that dominate real LLMs.

2. The experiments in this paper focus on synthetic tasks; it would be better if the authors could consider real-world language tasks.

**Questions:**

The authors claim that “without proper parameterization, models with larger expressive power surprisingly fail to generalize OOD after being trained by gradient descent.” However, real-world LLMs usually do not employ such specific parameterizations, then how do LLMs acquire their generalization ability?

---

> ### Author Response · Authors · 2025-11-21
>
> We thank the reviewer for the positive evaluation and feedback. Please find our answers below.
>
> > Limited architecture depth: Results are confined to one-layer, single-head transformers, far from the multi-layer, residual, or multi-head dynamics that dominate real LLMs.
>
> > The experiments in this paper focus on synthetic tasks; it would be better if the authors could consider real-world language tasks.
>
> Thank you for the suggestions. Since transformer theory (especially for multi-layer multi-head models) is an extremely challenging topic with no established frameworks for theoretical analyses, our main contributions on this work are intended to be focused on one-layer transformers on a well-defined task, in which the structural properties of the data samples are explicit. It is indeed a very interesting to extend the type of interpretable, human-like structures in our Section 4 and 5 to multi-head transformers for *multi-step* in-context recall tasks, we left this as a future work.
>
> Regarding the experiments: overall, we agree that an important future contribution would be having more experiments on reasoning datasets such as GSM8K.
> In the scope of our work, we chose the same synthetic setup that were used by the two prior papers on similar in-context recall tasks,  Bietti et al. (NeurIPS 2023) and Chen, Bruna and Bietti (ICLR 2025) to allow direct comparisons.
> Our Appendix A presents several real-world examples of in-context recall sentences for indirect object identification and transitive inference.
> To the best of our knowledge, there has not been any standard, well-established dataset for in-context recall tasks yet. Constructing such a dataset and testing larger models is an interesting future work.
>
>
>
> > The authors claim that “without proper parameterization, models with larger expressive power surprisingly fail to generalize OOD after being trained by gradient descent.” However, real-world LLMs usually do not employ such specific parameterizations, then how do LLMs acquire their generalization ability?
>
> We clarify that in this context, the "models with larger expressive power" part is referring to the non-parameterized one-layer transformers in our experiment section.
> These models, which are denoted by the prefix "Origin" in their names in our Table 1, have larger expression power than the parameterized models because all of their matrices are fully trainable, so their solution space encompasses the solution space of the parameterized models.
> We will soon upload a new version with this clarification.

---

### Official Review · Reviewer_4JXf · 2025-11-01

**Soundness:** 3
**Presentation:** 3
**Contribution:** 2
**Rating:** 4
**Confidence:** 4

**Summary:**

This paper provides a comprehensive analysis of the approximation, optimization, and generalization problem of in-context reasoning tasks with Transformers. The studied Transformers include linear, relu, and softmax attention. The generalization also involves the out-of-domain case on unseen data. Some experiments are provided to support the reparameterization considered in this work.

**Strengths:**

1. The setting of in-context reasoning is an important and interesting problem to study.

2. The analysis is comprehensive, which contains many aspects of the theory.

**Weaknesses:**

1. The analysis is simplified to consider only $\lambda$ as the trainable parameter. This is too restrictive.

2. The writing can be improved. Why not put real-world examples right after Definition 2.1?

3. The experiments only show the necessity of reparameterization. However, I think it only applies to synthetic experiments with two-layer models. It is clear whether reparameterization is important in real-world experiments.

**Questions:**

1. I am fine with experiments with synthetic data and settings. However, I feel experiments that can verify Theorems 5.4, 5.5, and Eqn. (4) are more interesting. I noticed the results of Figure 3. Why not put them in the main body?

2. Can your analysis and the results be extended to multi-head and/or multi-layer Transformers?

3. Why do your gradient updates need to be normalized?

4. How do you motivate the reparameterization with only $\lambda$ as the learnable parameter? Maybe you can cite some papers that support the formulation of $W$, e.g., some works [1, 2, 3, 4] show that attention scores are concentrated on tokens with the same feature as the query.

[1] Huang et al., ICML 2024. In-context convergence of transformers.

[2] Li et al., ICML 2024. How Do Nonlinear Transformers Learn and Generalize in In-Context Learning?

[3] Li et al., ICLR 2025. Training nonlinear transformers for chain-of-thought inference: A theoretical generalization analysis.

[4] Huang et al., ICLR 2025. A Theoretical Analysis of Self-Supervised Learning for Vision Transformers.

---

> ### Author Response · Authors · 2025-11-21
>
> We thank the reviewer for their constructive feedback and suggestions. Please find our answers below.
>
> > The writing can be improved. Why not put real-world examples right after Definition 2.1?
>
> Thank you for the suggestion. Due to the page limit for the submission, we were unable to put a lot of real-world examples in the main text. In the Introduction section, we specified a running example with the sentence “After talking to Bob about Anna, Charles gives
> her email address to [?]”. Our Appendix A presents several real-world examples of in-context recall sentences for indirect object identification and transitive inference. We will add these examples into the main text in the final version.
>
> > I noticed the results of Figure 3. Why not put them in the main body?
>
> We thank the reviewer for their high interests in our empirical results for the finite-sample analyses in our Section 5.
> Due to the page limit for the submission, we were unable to put Figure 3 and its analysis in the main text.
> We will add Figure 3 and its analysis into the main text in the final version.
>
> > Can your analysis and the results be extended to multi-head and/or multi-layer Transformers?
>
> Great question! The short answer is yes, we believe that the our proposed re-parameterization for one-step recall can be extended to multi-step recall tasks. It is likely that multi-step recall tasks will admit, as an interpretable parameterization solution, a re-parameterized multi-layer transformer with special structures that give rise to linear convergence rate and out-of-distribution generalization.
> However, we expect that analyzing such a multi-layer transformer is very challenging, especially with softmax attentions, and so this is best left for a future work.
>
> > Why do your gradient updates need to be normalized?
>
> This is also a good question. In the main text, line 231, we stated that the normalized gradient updates allow us to establish distribution-agnostic convergence results. Due to the space limit, we could not put more explanation in the main text for this part. In essence, the normalizing step allows us to reduce the training algorithm to be a scaled variant of sign-descent, where the scale factor is $\frac{1}{Q}$. This alleviates the need for knowing the conditional distribution of sentences given a pair of (trigger, output) tokens. This normalization is especially helpful in the two-phase analysis of the models with softmax attention, since it allows us to control the number of time steps in Phase 1, where the signs of the updates fluctuate.
>
> > How do you motivate the reparameterization with only $\lambda$ as the learnable parameter? Maybe you can cite some papers that support the formulation of , e.g., some works [1, 2, 3, 4] show that attention scores are concentrated on tokens with the same feature as the query.
>
> Thanks for the suggested references, our paper already cited some of them and will add the rest in the next version. In essence, our reparameterization, especially for the attention circuit, is motivated from simulating how humans would actually solve the noiseless instance of this in-context recall task.  We already mentioned that our reparameterization derives a human-like rule for solving this task, but we will emphasize this point more in an updated version (to be uploaded soon).

---

### Official Review · Reviewer_aJo2 · 2025-11-04

**Soundness:** 1
**Presentation:** 2
**Contribution:** 2
**Rating:** 4
**Confidence:** 4

**Summary:**

The paper studies a stylized in-context recall task and trains a one-layer decoder transformer with linear/ReLU/softmax attention under a reparameterization. It proves Bayes-optimality of the construction (noiseless and noisy), linear-rate convergence under normalized GD, and OOD generalization to unseen output words; it also gives a finite-sample guarantee and a result showing attention vs. FFN role separation at convergence.

**Strengths:**

- Clear theoretical guarantees across three attention types. Linear/ReLU (Lemma 3.1, Thm. 3.2) and softmax (Lemma 4.1, Thm. 4.2) get explicit parameterizations with linear convergence proofs.

- OOD to unseen outputs is formalized and proved in both noiseless and noisy settings (Thm. 3.3, Thm. 5.5).

- Mechanistic interpretability hook: theorem showing attention predicts outputs while FFN handles noise after enough steps (Thm. 5.6).

**Weaknesses:**

1. The abstract states: Existing theoretical results only focus on the in-context reasoning
behavior of transformers after being trained for the one gradient descent step.
This is not correct. Several papers analyze full training dynamics over many GD steps and prove convergence (often linear/finite‐time), not merely “one step” (you cited the first one, and didn't cite the last two), e.g.:

[1] Huang, Cheng & Liang (2023). In-context convergence of transformers. and In-context learning with representations: Contextual generalization of trained transformers.

[2] Yang, Huang, Liang & Chi (2024). In-context learning with representations: Contextual generalization of trained transformers.

[3] Shen, Zhou, Yang, Shen (2025). On the Training Convergence of Transformers for In-Context Classification of Gaussian Mixtures.
...

---

2. It seems to me that there are some quantitative mismatch that makes some theorem statements numerically wrong (too optimistic) in their dependence on $|Q|$ and $t$:/

- Theorem 3.2 (NGD dynamics): Main text states $\lambda_{q,t} = \eta t/|Q|$ and uses $L(\lambda_t)=O(Ne^{-\eta t})$. But the appendix’s NGD derivation (equal gradient components $\Roghtarrow ||\nabla L||_2 = \sqrt{|Q|}|\partial L/\partial \lambda_q|) gives $\lambda_{1,t}=\eta t/\sqrt{Q}$. Elsewhere you even plug $\lambda_{q,t}=\eta t$. So the printed $|Q|$ and $\eta t$ versions overstate how fast $\lambda$ grows and thus how fast the loss decays.

- In Theorem 3.3, The lower bound that uses $exp(\eta t)$ should be using $\exp(\eta t/|Q|)$. As printed, it predicts higher accuracy sooner.

- Theorem 4.2, a softmax loss bound is written as $O(Ne^{-t})$, but it should retain $\eta: O(Ne^{-\eta t})$.

3. The task seems a bit artificial to me: The vocabulary is partitioned into trigger tokens $Q$ and output tokens $O$, plus a single “generic noise” token $\tau$. Sentences are forced to contain at least one $(q,y)$ bigram; any $(q,x)$ bigram must have $x\in\{y,\tau\}; and the final position is fixed to the trigger $z_H=1$. This raises SNR but is artificial as a language model training distribution.

**Questions:**

1. Can you compare with the missing literature? What are the novelty/contribution compared to them? e.g., the attention machanism seems very similar to Huang et al ([1], you both consider analyzing one matrix W, and for the softmax version, it's the same as Huang et al and converges to $s I$ with $s\rightarrow \infty$).

---

> ### Author Response · Authors · 2025-11-21
>
> We thank the reviewer for their constructive feedback and suggestions. Please find our answers below.
>
> > The abstract states: Existing theoretical results only focus on the in-context reasoning behavior of transformers after being trained for the one gradient descent step. This is not correct. Several papers analyze full training dynamics over many GD steps and prove convergence (often linear/finite‐time), not merely “one step” (you cited the first one, and didn't cite the last two), e.g.: [1], [2], [3]
>
> We clarify that in this context, the term "in-context reasoning" is taken from Bietti et al. (NeurIPS 2023) paper, which refers to the in-sentence-context object recall task only.
> For this in-context recall tasks, the two existing papers with substantial theoretical claims are Bietti et al. (NeurIPS 2023) and Chen, Bruna and Bietti (ICLR 2025), both of which consider only the one gradient descent step right after initialization.
> We did realize the possible confusion in characterizing this task as "in-context reasoning", and hence we have tried to use the term "in-context recall" throughout our paper instead. We will soon upload a new version with a clearer abstract, as well as add the aforementioned references in the related work section. Please note that the references [1], [2], [3] suggested by the reviewer are for regression tasks, which are fundamentally different from next-token prediction tasks.
>
> > It seems to me that there are some quantitative mismatch that makes some theorem statements numerically wrong (too optimistic) in their dependence on $|Q|$ and $t$
>
> Thanks for pointing out these source of confusions in our writing. For our theoretical results, we always state the full statements, either in the appendix or the main text, and then present a simplified version to highlight the dependency on a certain quantity.
> Our intention was to improve readability, however we admit that this might have caused more confusion that we expected.
> We will soon upload a new version with a clearer presentation on these results.
>
> > Can you compare with the missing literature? What are the novelty/contribution compared to them? e.g., the attention machanism seems very similar to Huang et al ([1], you both consider analyzing one matrix W, and for the softmax version, it's the same as Huang et al and converges to $sI$ with $s \to \infty$
>
> This is actually a two-part question. We answer each of them below.
>
> On missing literature: first, we would like to clarify that the three references provided by the reviewer are studies on *regression tasks with squared loss*, which are fundamentally different from next-token prediction (NTP) tasks. Several major differences are the range of the expected outputs (real-output versus categorical output), loss functions (square versus logistic), input encoding (feature vectors versus positional embedding). Additionally, analyzing NTP tasks with softmax attention often require taking derivatives of a loss function of nested softmax operators, which are generally more challenging to work with than square loss of one-level softmax.
>
> On attention mechanism in comparison to Huang et al. [1]: there are two fundamental differences.
> - Conceptual difference: because the task in Huang et al. is regression, their attention score of to a particular token in a prompt does not depend on the position of the token. In contrast, as we have emphasized throughout our paper (e.g. in the abstract, and in the attention score formula on line 212 and 213 in Section 3), in our NTP task, a crucial property is that the positional information of a token largely determine its attention score.
>
> - Technical difference: the value and key-query matrices in our work have very different structure than those in Huang et al. (i.e. their Equation 2 on page 4 at https://arxiv.org/pdf/2310.05249). Their value matrix $W^V$ is fixed to have zeros everywhere except for the last entry of the diagonal, whose value is fixed to be $\nu = 1$. In contrast, our value matrix $V$ is a scaled identity matrix, and this scale factor is trainable for softmax attention. Furthermore, their key-query matrices $W^{KQ}$ contain only one non-zero diagonal block. This is very different from our key-query matrix $W$, which takes the form of an associative memory between trigger tokens, output tokens and noise tokens (our Lemma 3.1, 4.1 and 5.1).

---

### Author Response · Authors · 2025-11-25
**New Revision Uploaded**

Dear Reviewers,

We have uploaded a revision that addresses all of your feedback. The changes are marked by green color. We emphasize that all of these changes are meant to improve clarity of our work, and do not alter our contributions in any way. Please let us know if your have other comments or feedback.

The list of changes are below:
- Abstract: the term "in-context reasoning" has been changed to "in-context recall" for better clarity on the scope of our setup.
- Abstract: replaced "models with larger expression power" by "standard one-layer transformers" for better clarity on the comparison models.
- Line 155: added a sentence emphasizing that previous work did not study OOD generalization for in-context recall tasks
- Line 207: added references for prior work that used re-parameterized matrices
- Theorem 3.2, 3.3, 4.2: added $1/|Q|$ back to the bound (does not impact the convergence rates)
- Appendix: added Appendix B for a summary of differences between the setup and technical mechanism of our linear convergence rates versus those in Huang et al. (NeurIPS 2024)
- Appendix: added Appendix C for a summary of differences between our analysis and results versus those in Bietti et al. (NeurIPS 2023)
- Throughout the paper: fixed minor typos and grammar inaccuracies
---
Huang et al. (NeurIPS 2024). Non-asymptotic Convergence of Training Transformers for Next-token Prediction.

Bietti et al. (NeurIPS 2023). Birth of a transformer: A memory viewpoint.

---

### Meta-Review · Area_Chair_np6c · 2026-01-01

**Summary:**

This paper considers a one-layer decoder-only transformer trained on an in-context recall task, and it proves results about (i) approximation capability, (ii) convergence of gradient descent, and (iii) generalization error. The model is a variation over earlier work by (Bietti et al., 2023; Chen et al., 2025). The manuscript starts by showing the approximation capabilities, the convergence rate of gradient descent, its OOD generalization, and its implicit bias for transformers with linear and ReLU attention in a noiseless setting. Next, it shows that the population loss converges to 0 at a linear rate for softmax attention. Finally, it provides results in the noisy setting (approximation capability, training dynamics and generalization for finitely-many samples).

Overall, the reviewers have appreciated the clear theoretical guarantees across different attention types (Linear, ReLU and softmax) and setups (noiseless and noisy), and the interesting problem being tackled which extends earlier work limited to a single gradient step. The reviewers have identified possibly related works which have been then discussed by the authors in a thoughtful rebuttal. However, some limitations remain, mostly related to the over-simplified setting (only $\lambda$ trainable; only one-layer single-head architectures considered; somewhat artificial task). Thus, I am unable to accept the manuscript in its present form and, while I think that the paper has some clear strengths, I believe that an additional round of reviewing is warranted before it can be accepted.

**Reviewer Concerns:**

Reviewer aJo2. The connection with the results mentioned by the reviewer has been clarified in a satisfactory manner. The issues related to the dependence on $|Q|$ and $t$ have been fixed as well. The concern related to the fact that the task is a bit artificial remains outstanding.

Reviewer 4JXf. The issues related to the model being restrictive (weakness 1, question 2) remain outstanding. The other questions have been properly addressed.

Reviewer xnEr. There was no substantial issue and the reviewer was borderline positive.

Reviewer e9iq. The reviewer mentions several works that could potentially reduce the novelty of the findings, but the authors argue that such papers are either not related or not solving the same problem. Here, a longer discussion period would have been beneficial. However, even discarding this concern, I find it rather unlikely that a consensus towards accepting the paper could have been reached.

**Reviewer Scores:**

While it is possible that some scores would have changed if the reviewers had been able to participate fully in the discussion, I still find it unlikely that a consensus towards accepting the paper would have been reached.

---

### Decision · Program_Chairs · 2026-01-26

Reject